# Tumor acidosis-induced DNA damage response and tetraploidy enhance sensitivity to ATM and ATR inhibitors

Léo Aubert [1]✉, Estelle Bastien [1], Ophélie Renoult [1], Céline Guilbaud[1], Kübra Özkan[1], Davide Brusa [2], Caroline Bouzin[3], Elena Richiardone[1], Corentin Richard[4], Romain Boidot [4], Daniel Léonard [5], Cyril Corbet [1] & Olivier Feron [1,6]✉

## Abstract

**Tumor acidosis is associated with increased invasiveness and drug resistance. Here, we take an unbiased approach to identify vulnerabilities of acid-exposed cancer cells by combining pH-dependent flow cytometry cell sorting from 3D colorectal tumor spheroids and transcriptomic profiling. Besides metabolic rewiring, we identify an increase in tetraploid cell frequency and DNA damage response as consistent hallmarks of acid-exposed cancer cells, supported by the activation of ATM and ATR signaling pathways. We find that regardless of the cell replication error status, both ATM and ATR inhibitors exert preferential growth inhibitory effects on acid-exposed cancer cells. The efficacy of a combination of these drugs with 5-FU is further documented in 3D spheroids as well as in patient-derived colorectal tumor organoids. These data position tumor acidosis as a revelator of the therapeutic potential of DNA repair blockers and as an attractive clinical biomarker to predict the response to a combination with chemotherapy.**

**Keywords** Tumor Acidosis; 3D Spheroids; DNA Damage Response; ATM; Organoids
**Subject Categories** Cancer; DNA Replication, Recombination & Repair; Metabolism

## Introduction

Several discoveries have shed light on the extracellular acidic tumor microenvironment reported three decades ago in human tumors *vs.* physiological extracellular pH ($pH_e$) measured in surrounding healthy tissues (Griffiths, 1991; Vaupel et al, 1989). This original observation has indeed sparked interest in understanding how solid tumors could take advantage of extracellular acidic pH values (Corbet and Feron, 2017; Ibrahim-Hashim and Estrella, 2019; Pillai et al, 2019). In parallel, investigators developed methods that can provide tumor $pH_e$ measurements with spatial resolution (Anemone et al, 2019) and explored the therapeutic potential of targeting cancer cells located in the acidotic tumor compartment (termed *acid-exposed cancer cells* here below) (Blaszczak and Swietach, 2021; Deskeuvre et al, 2022; Dierge et al, 2021; Kolosenko et al, 2017; Parks et al, 2013). Today strong evidence indicate that extracellular tumor acidic conditions are associated with metastatic spreading and therapeutic resistance, and thus largely contribute to cancer cell aggressiveness (Corbet et al, 2020; Rohani et al, 2019; Wojtkowiak et al, 2011; Yao et al, 2020).

The acidic tumor microenvironment, often referred to as tumor acidosis, is a consequence of the increased metabolic activity of cancer cells, which leads to the accumulation of protons, either as a direct waste product from metabolic reactions or through hydration of $CO_2$ released by oxidative cancer cells (Corbet and Feron, 2017; Ibrahim-Hashim and Estrella, 2019; Pillai et al, 2019). Importantly, this latter contribution of mitochondrial activity to acidosis also means that acidosis and hypoxia do not fully overlap (Corbet and Feron, 2017). The compromised tumor vasculature further contributes to a deficiency in removing $H^+$ from the milieu bathing some tumor areas. Cancer cells located in acidic areas of primary tumors are thus distinct from the highly proliferating cancer cells usually located in the more physiological (buffered) environment nearby perfused blood vessels (Corbet and Feron, 2017). In the perspective to identify anticancer interventions prone to reach these more quiescent acid-exposed cancer cells, 3D tumor spheroids represent a unique platform to get access to an in vitro model wherein a $pH_e$ gradient develops spontaneously with the progressive formation of an acidic compartment.

[1]Pole of Pharmacology and Therapeutics (FATH), Institut de Recherche Expérimentale et Clinique (IREC), UCLouvain, B-1200 Brussels, Belgium. [2]CytoFlux-Flow Cytometry and Cell Sorting Platform, Institut de Recherche Expérimentale et Clinique (IREC), UCLouvain, B-1200 Brussels, Belgium. [3]Imaging Platform 2IP, Institut de Recherche Expérimentale et Clinique (IREC), UCLouvain, B-1200 Brussels, Belgium. [4]Unit of Molecular Biology, Department of Biology and Pathology of Tumors, Georges-François Leclerc Cancer Center-UNICANCER, 21079 Dijon, France. [5]Institut Roi Albert II, Department of Digestive Surgery, Cliniques Universitaires St-Luc, and Institut de Recherche Expérimentale et Clinique (IREC), UCLouvain, B-1200 Brussels, Belgium. [6]Walloon Excellence in Life Sciences and Biotechnology (WELBIO) Department, WEL Research Institute, B-1300 Wavre, Belgium. ✉E-mail: leo.aubert@uclouvain.be; olivier.feron@uclouvain.be

Herein, we aimed to take an unbiased comprehensive approach to identify potential paths to be targeted in order to kill cancer cells located in the tumor acidic compartment. The use of pH (low) insertion peptides (pHLIP) has recently emerged as a technique to label acid-exposed cancer cells (Andreev et al, 2007; Rohani et al, 2019; Weerakkody et al, 2013). pHLIP peptides are membrane-inserting peptides exhibiting a remarkable specificity for acidic $pH_e$. We reasoned that by combining the unique targeting ability of pHLIP peptide with the high-resolution phenotypic analysis enabled by FACS sorting and RNA-based transcriptomic profiling, we could gain valuable insights into specific traits of the acid-exposed tumor cell population. By profiling gene expression in isolated acid-exposed cancer cells from 3D spheroids, we found enriched patterns supporting fatty acid (FA) metabolism and OXPHOS (Corbet et al, 2016; Michl et al, 2022; Rolver et al, 2023; Yao et al, 2020) but also identified pathways that had never been yet the subject of detailed investigation. Among them, DNA damage response (DDR) resonated with the increased aggressiveness of acid-exposed tumor cells. Indeed, through a myriad of complex molecular events to sense, repair, and mitigate the consequences of DNA lesions, DDR pathways intersect with several key cellular processes critical for cancer progression, including drug resistance and metastases formation (Aricthota et al, 2022; Groelly et al, 2023; Jeggo et al, 2016; Nickoloff, 2022). The observed DDR response is yet more relevant that we and others have previously reported that upon progressive adaptation of cancer cells to moderate extracellular acidosis (−0.5 to −0.9 pH unit vs. physiological pH 7.4), intracellular pH ($pH_i$) actually reaches slightly alkaline values (Corbet et al, 2014; Persi et al, 2018; White et al, 2017). Mild extracellular acidosis is thus not associated with long-term protonation of cytosolic or nuclear entities as was likely to happen in previous studies reporting the clastogenic effects of acidosis at clamped $pH_e$ values as low as 5.5–6 (Morita et al, 1992).

DDR components include ataxia telangiectasia mutated (ATM) and ataxia telangiectasia and Rad3-related (ATR) kinases (Groelly et al, 2023; Pilie et al, 2019). ATM primarily responds to double-strand breaks (DSBs), while ATR is activated by a wide range of DNA lesions, including single-strand breaks and stalled replication forks (Marechal and Zou, 2013). Upon activation, ATM and ATR initiate signaling cascades that lead to cell cycle arrest, DNA repair, and, when necessary, the induction of programmed cell death. ATM and ATR are induced in cancer cells exposed to chemotherapy and blocking them is proposed as an attractive strategy to overcome resistance in cancer patients (Nickoloff et al, 2017; Pilie et al, 2019).

In this study, the combination of pHLIP peptide with the high-resolution phenotypic analysis enabled by FACS and transcriptomic profiling provided by RNA-Seq offered an innovative platform for the identification of a strong DDR phenotype and tetraploidy in acid-exposed cancer cells. This led us to unravel that ATM and ATR inhibitors represent a strategy particularly suited to jeopardize DDR in acid-exposed cancer cells and exert growth inhibitory effects on this aggressive tumor compartment either as a single agent or in combination with chemotherapy. Synthetic lethality resulting from combining ATM or ATR inhibition with the increased susceptibility of acid-exposed cancer cells to undergo DNA damages positions tumor acidosis as an attractive predictive biomarker for the clinical use of these drugs. Altogether, these data bring a new stone to the edifice of the critical role of tumor acidosis and how targeting it may unveil novel opportunities to improve cancer patient outcomes.

# Results

## pHLIP-based strategy to isolate and characterize acid-exposed cancer cells from 3D spheroids

To get insights on potential therapeutic strategies to target acid-exposed cancer cells, we sought to identify transcriptome changes induced by the ambient acidic $pH_e$ developing in 3D colorectal cancer (CRC) spheroids. We used a pHLIP peptide conjugated to Alexa 568 fluorophore to identify the acidic compartment of 3D spheroids (Weerakkody et al, 2013) (Fig. 1A). When facing acidic $pH_e$, pHLIP peptide folds into a coil conformation which inserts into cell membranes and thus allows selective labeling of acid-exposed cancer cells (pH 6.0–6.8). HCT116 CRC cells were used for their ability to form highly reproducible large spheroids largely deprived of a necrotic zone prone to interfere with labeling. In our hands, the use of pHLIP peptides led to the labeling of the center of the spheroids but also of the rim of the 3D structures as revealed on equatorial sections of spheroids (Fig. 1B, top left panel). The peripheral staining at the spheroid/medium interface evoked unspecific labeling due to membrane adsorption without the expected coil-helix transition and membrane integration occurring at low $pH_e$. To prove this issue, we used a K-pHLIP peptide wherein protonatable aspartate residues are replaced by positively charged lysine residues so that it loses its ability to fold and integrate into plasma membranes of cells exposed to acidic extracellular pH (Weerakkody et al, 2013). This K-pHLIP control experiment confirmed the unspecific (ie, non-pH-dependent) peptide labeling of the external layers of 3D spheroids (that are in close contact with the buffered culture medium) (Fig. 1B, top right panel). Fluorescence measurements revealed a similar signal extent of pHLIP and K-pHLIP peptides at a depth <50 μm, this level was higher than the detected signal in the underlying cell layers (between 50 and 100 μm) but much lower than the central acidic core of the spheroids only labeled by pHLIP peptide (Fig. 1B, lower panel). We therefore sorted acid-exposed cancer cells from 3D spheroids using both positive pHLIP and negative K-pHLIP labeling (Figs. 1C,D and EV1A,B) and performed genome-wide RNA-sequencing (RNA-seq) analysis. Here below, we will thus use the terms *acid-exposed cancer cells* to refer to cancer cells exposed to a low extracellular pH that we will compare to the pHLIP-negative cell population that we will name *non-acid-exposed cancer cells*.

RNA-seq analysis of consistent biological replicates (validated by PCA analysis (Fig. EV1C)) led to the identification of ≈17% (#3550) of expressed genes and ≈15% (#3026) of protein-coding genes differentially expressed (DE) in response to acidosis (TPM > 0.5 and P value < 0.05) (Fig. EV1D,E; Dataset EV1). Nonetheless, only 144 and 64 protein-coding genes were significantly upregulated and downregulated, respectively, when we considered P value < 0.05 and $Log_2$ fold change (FC) [acid/non-acid] ≥ |1| (Figs. 1E and EV1D). Gene Set Enrichment Analysis (GSEA) revealed that upregulated genes were enriched for the oxidative phosphorylation, fatty acid metabolism and reactive oxygen species pathways, which have been previously identified by us and others as specific features of cancer cells adapted to acidic $pH_e$ (Corbet et al, 2016; Michl et al, 2022; Rolver et al, 2023; Yao et al, 2020) (Fig. 1F).

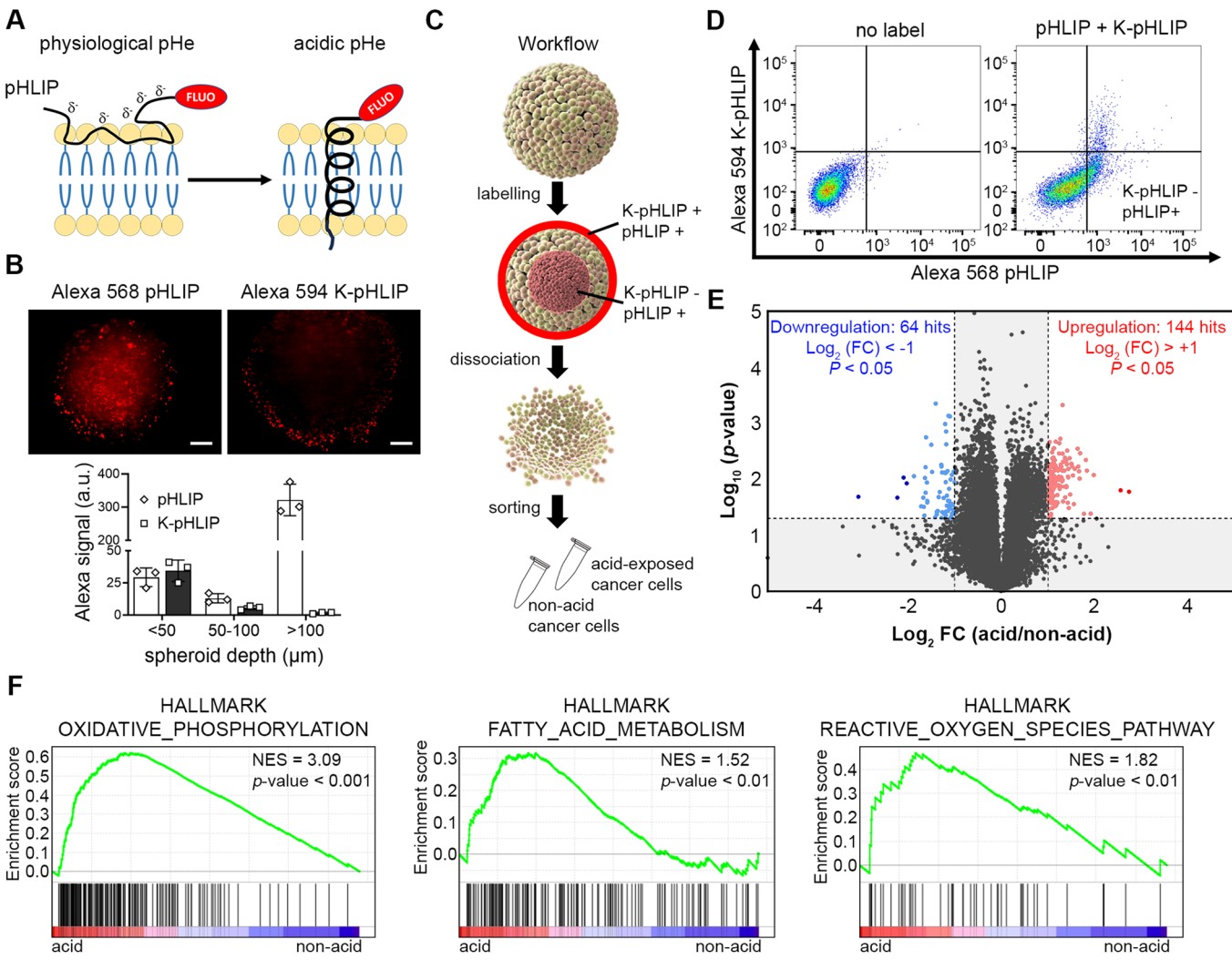

**Figure 1. pHLIP-based sorting of acid-exposed cancer cells from 3D tumor spheroids and transcriptomic validation.**

(A) Diagram depicting the molecular mechanism for spontaneous folding and insertion of the pH (low) insertion peptide (pHLIP) into the cell membrane of acid-exposed cancer cells. (B) Labeling of a 3D HCT116 spheroid equatorial section with either pH-dependent Alexa 568 pHLIP or pH-independent Alexa 594 K-pHLIP (negative control). Scale bars = 100 μm. (C) Schematic representation of the Fluorescence-activated cell sorting (FACS) approach used to isolate pHLIP-positive, K-pHLIP-negative acid-exposed cancer cells (*vs.* pHLIP-negative non-acid cancer cells) from 3D spheroids. (D) FACS density plots depicting Alexa 568 pHLIP *vs.* Alexa 594 K-pHLIP fluorescence intensity for unlabeled (left side) and double-labeled (right side) cells sorted from 3D HCT116 spheroid. (E) Volcano plot of differentially expressed protein-coding genes between isolated acid-exposed and non-acid HCT116 cancer cell populations. Shaded areas represent the cutoff range of negative $\log_{10}$ (P value) *vs.* $\log_2$ fold change [FC] (acid/non-acid). $\log_2$ FC (acid/non-acid) above 1 or below −1 (i.e., twofold changes) and P values ≤ 0.05 were considered as significantly upregulated (red) or downregulated (blue), respectively. (F) GSEA plots showing enrichment in hallmark gene sets of oxidative phosphorylation, fatty acid metabolism and reactive oxygen species pathway in acid-exposed *vs.* non-acid cells. Data information: (B) quantification data (bar graph, below) are presented as means ± SD of $n = 3$ independent biological replicates. (E, F) Data are represented as mean of $n = 3$ independent biological replicates and P values were calculated by DESeq2 with Benjamini–Hochberg multiple test correction (E). GSEA normalized enrichment scores (NES) and P values are indicated in (F). Source data are available online for this figure.

Thus, gene enrichment data analysis together with the methodological precautions emphasized above validate pHLIP-based strategy to isolate and characterize acid-exposed cancer cells arising from the spontaneous formation of a $pH_e$ gradient in 3D tumor spheroids.

## The acidotic compartment of 3D spheroids exhibits a robust DDR signature

To further elucidate specific traits of acid-exposed cancer cells that could represent druggable pathways, we conducted a comprehensive pathway and process enrichment analysis of the 208 DE protein-coding genes by using the Metascape platform (Zhou et al, 2019) (Fig. 2A). Furthermore, we performed a network analysis to better capture the relationships between the enriched pathways, where each node represents an enriched term and is colored by its cluster ID (Fig. 2B) or by its P value (Fig. EV2A). We found that DNA repair and several cell cycle-related pathways (i.e., mitotic cell cycle, regulation of nuclear division, retinoblastoma gene, meiotic cell cycle) were among the most enriched functional clusters in the acid-associated transcriptome (Figs. 2A,B and EV2A). Furthermore, the other top-listed functional clusters

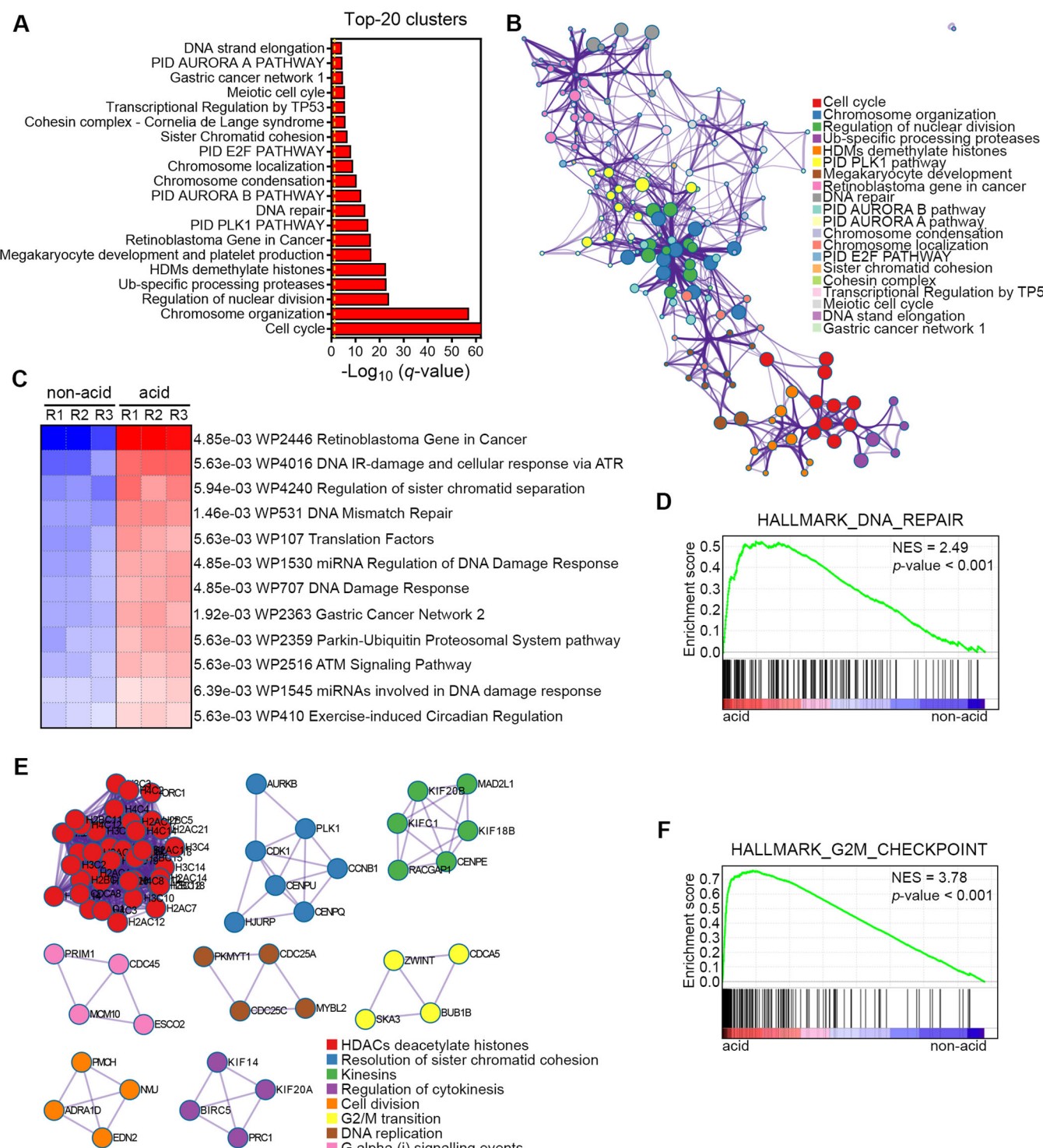

(e.g., HDMs demethylate histones, sister chromatid cohesion, TP53 activity, PLK-1 pathway, or AURORA B) have also been linked to one or several critical steps in the DDR (Ferrari and Gentili, 2016; Gong and Miller, 2019). Strikingly, most of these enrichments were almost exclusively due to the 144 upregulated DEGs (Fig. EV2B). In line with the Metascape analysis, we then performed a gene-annotation enrichment of the 144 upregulated DEGs using

g:Profiler, and found that most significant enrichments in GO:CC and Reactome terms were associated either with chromosome dynamics or cell cycle checkpoints which both resonate with DNA damage-induced chromatin remodeling (Fig. EV2C,D). Based on GSEA and parametric GSEA (PGSEA), we further confirmed that ambient acidic pH significantly increased transcripts associated with *DNA repair hallmark* and many DDR-related pathways

**Figure 2. Acid-exposed cancer cells in 3D tumor spheroids exhibit a DNA damage response (DDR) signature and enrichment in genes related to G2/M checkpoint.**

(A, B) Bar chart (A) and network plot (B) of the top-20 most significant Metascape-annotated functional clusters enrichment correlating with differentially expressed protein-coding genes from the transcriptome of acid-exposed cancer cells isolated from 3D spheroids. (B) Each circle node represents a distinct pathway annotation colored according to its Metascape-identified cluster and with a size proportional to the number of pathway-associated genes. The thickness of the purple edges indicates the number of common genes between various pathway annotations. (C, D) PGSEA (C) and GSEA (D) analyses using WikiPathways and Hallmark databases, respectively, revealed a propensity of transcripts of acid-exposed cancer cells to positive enrichment in DDR-related pathways. (E) Protein–protein interactions (PPI) network analysis of the DE protein-coding genes in acid-exposed cancer cells using the Metascape MCODE algorithm to identify neighborhoods where proteins are densely connected. Each MCODE network corresponds to the most significant enriched pathway annotation and is assigned a unique color. (F) GSEA analysis revealed that HALLMARK_G2M_CHECKPOINT gene set is positively enriched in acid-exposed cancer cells. Data information: (A, B) significance was determined using multi-test adjusted $P$ values ($q$ value) in negative $\log_{10}$ ($Q < 0.05$). GSEA normalized enrichment scores (NES) and $P$ values were used in (D, F).

(i.e., DNA Mismatch Repair, ATM and ATR signaling pathways) from Wikipathways (WP) and Pathway Interaction Database (PID) (false-discovery rate (FDR) < 0.1) (Figs. 2C,D and EV2E). Accordingly, we also noticed that 15% (#88) of genes encompassing the DNA Repair Gene Ontology (GO0006281) term in the acid-associated transcriptome were upregulated (Log$_2$ FC (acid/non-acid) > |0.5| and $P$ value < 0.05) while less than 1% (#5) were downregulated (Fig. EV2F). Protein–protein interaction (PPI) enrichment analysis using the Molecular Complex Detection (MCODE) algorithm from the Metascape tool, also identified eight significant densely connected network components, including a significant hub related to histone deacetylases (HDACs), well-known actors in DDR pathways (Roos and Krumm, 2016) and other subnetworks associated with cell cycle progression (Figs. 2E and EV2G). The latter observation was supported by GSEA that revealed significant increase of the *G2/M pathway*, further highlighting a link between acid-upregulated transcripts and DNA damage checkpoints (Fig. 2F).

## Acid-exposed cancer cells undergo cycle arrest in the G2/M phase and tetraploidy together with a strong activation of ATM and ATR pathways

Next, we performed cell cycle analysis on HCT116 colorectal cancer cells chronically adapted to acidic pH$_e$ 6.5 *vs.* pH$_e$ 7.4 as previously described (Corbet et al, 2020; Corbet et al, 2014; Corbet et al, 2016). Interestingly, we found a dramatic increase in the amounts of acid-exposed cancer cells in G2/M phase, strongly suggesting that HCT116 cells cultured at pH 6.5 were more prone to cell cycle arrest (Fig. 3A). In addition, flow cytometric analysis of DNA content from acidic pH-adapted cancer cells also demonstrated distinct PI staining in the tetraploid region (Fig. EV3A) that corresponds to a >20-fold increase of tetraploid cells (Fig. 3B), strongly supporting the emergence of genomic instability at acidic pH$_e$. Although we cannot formally exclude that at least part of the diploid G2/M signal is contaminated by tetraploid G1 phase cancer cells that share a theoretical 4 N DNA content (see also Fig. EV3 legend), the G2/M arrest is in line with our GSEA identifying a significant increase of the G2/M pathway in acid-exposed cancer cells (see Fig. 2F).

HCT116 is a K-RAS mutated CRC cell line that is DNA replication error (RER)-positive because of the lack of *hMHL1* expression, a central actor of the DNA mismatch repair (MMR) pathway (Kennedy et al, 2000). To determine whether the observed increase in tetraploidy was related to MMR defects and mutation accumulation, we next aimed to examine whether acidosis could similarly promote DDR in the RER-negative (K-RAS wild-type) CRC cell line HT-29 (Bracht et al, 2010). Accordingly, under acidic conditions, a net increase in the proportion of HT-29 cells in G2/M

phase was also observed, together with a significant increase in tetraploid cells (Fig. EV3B–D). We also took advantage of acidosis developing in proportion to the size of 3D tumor spheroids as previously reported (Corbet et al, 2020), and documented an increase in the number of tetraploid cells in HT-29 spheroids with a diameter >500 μm (*vs.* those with a diameter <300 μm) (Fig. EV3E).

We then reasoned that as observed in the acidic compartment of 3D spheroids, G2/M arrest was very likely to be associated with pathways involving main players in the DDR, namely the kinases ATM and ATR, which are primarily activated in response to DSBs and stalling or slowing of DNA replication as observed upon single-strand breaks (SSBs), respectively (Pilie et al, 2019). We found that most genes contributing to both pathways were significantly upregulated in the pHLIP-positive acidic spheroid compartment, supporting an acidosis-mediated increase in sensing DNA damages (Fig. 3C,D). Altogether, these findings pinpoint DDR, G2/M cell cycle arrest and tetraploidy as robust characteristics of acid-exposed cancer cells.

## Acid-exposed cancer cells exhibit activation of ATM, ATR, and CHK1/2 kinases similar to that of non-acid cancer cells exposed to chemotherapy

To further investigate the possibility to take advantage of DDR to target acid-exposed cancer cells, we next examined the phosphorylation of ATM and ATR, and downstream checkpoint kinases CHK1 and CHK2 in colorectal cancer HCT116 cells cultured at pH$_e$ 6.5 or pH$_e$ 7.4. Cancer cells chronically adapted to pH$_e$ 6.5 revealed a significant increase in phospho-ATM and phospho-ATR together with higher phospho-CHK1 and phospho-CHK2 levels (*vs.* cells maintained at pH$_e$ 7.4) (Fig. 4A). An acute pH$_e$ titration of the medium bathing native HCT116 cancer cells confirmed a pH-dependent elevation in phospho-CHK1 levels; phospho-CHK2 was also increased upon acute exposure to pH$_e$ 6.5 but not at lower pH$_e$ probably because of the development of cytotoxicity at pH$_e$ < 6.0 (Fig. 4B). Remarkably, similar increases in phospho-CHK1 and -CHK2 were observed in RER-negative CRC HT-29 cells as well as in a cancer cell line of distinct origin, namely cervix SiHa cancer cells (Fig. 4C). These data identify acidosis as an environmental condition prone to stimulate DDR regardless of the susceptibility to replication error (e.g., MMR deficiency).

To further support the activation of ATM and ATR pathways under acidosis, we examined them in cancer cells exposed to genotoxic 5-fluorouracil (5-FU). While the proportion of acid-exposed cells in the G2/M phase did not further increase upon exposure to 1 μM 5-FU (Fig. EV4A), the number of tetraploid cells significantly increased up to 30% of total acid-exposed cancer cells (*vs.* ~1% of 5-FU-treated cancer cells at pH$_e$ 7.4) (Fig. 4D). We also found that 5-FU-treated RER-positive HCT116 cancer cells (at pH$_e$ 7.4) exhibited a dose-dependent increase in

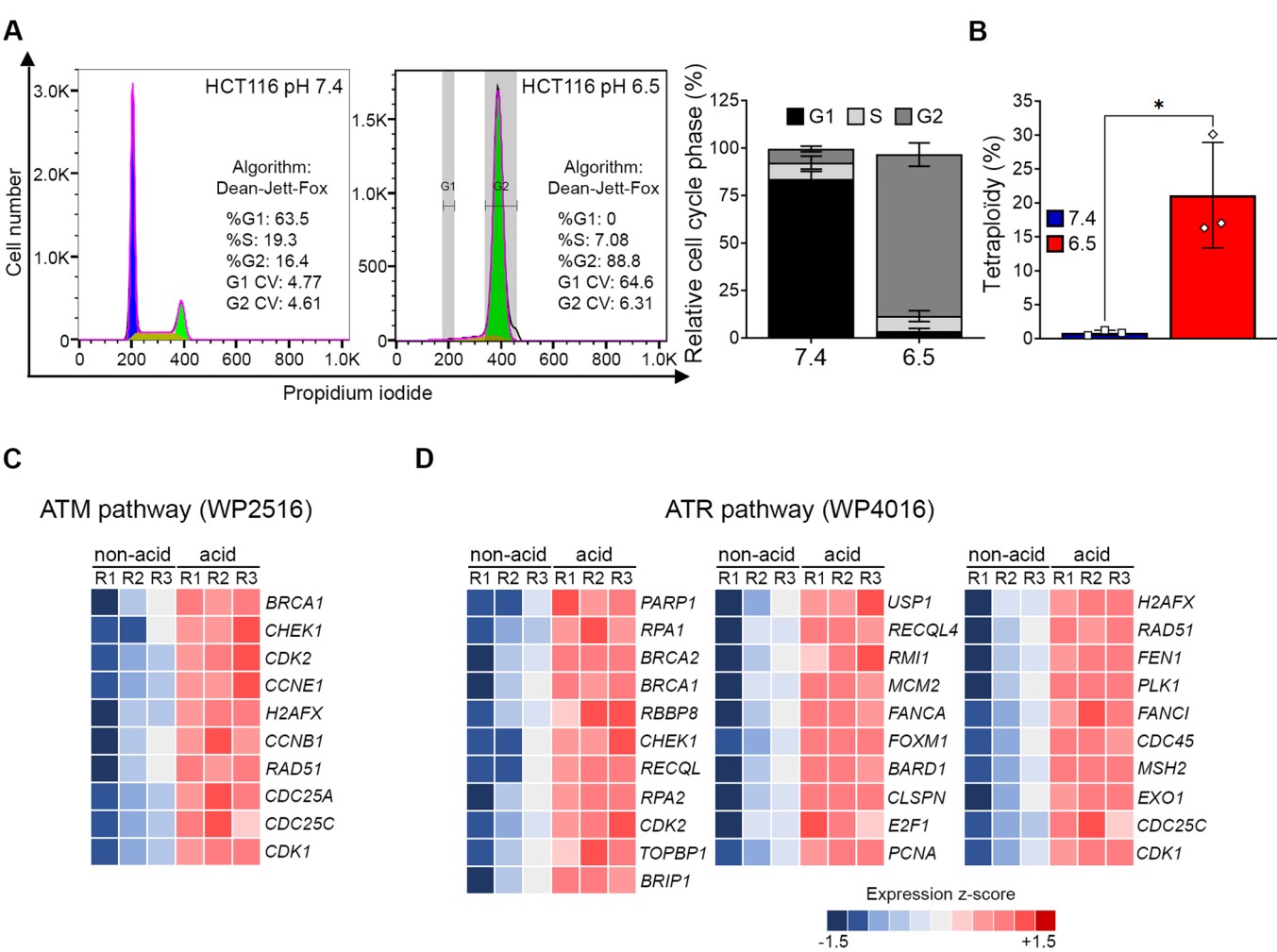

**Figure 3. Acid-exposed cancer cells accumulate in G2/M cycle phase and activate ATM/ATR pathways.**

(A) Flow cytometry analysis of DNA content was used to determine the cell cycle distribution of HCT116 cells cultured at pH 6.5 vs. pH 7.4. Representative cell cycle analysis (A, left panels) and quantification (A, right bar graph) are shown. (B) Proportion of tetraploid HCT116 cancer cells determined as described in Fig. EV3A from cells cultured at pH 6.5 or 7.4. (C, D) Heatmap of relative expression of genes (Log$_2$ FC (acid/non-acid) >|0.5| and $P < 0.05$) that are involved either in (C) ATM pathway (WP2516) or (D) ATR pathway (WP4016). Data information: (A, B) quantification data are presented as means ± SD of $n = 3$ independent biological replicates. The relative proportion of HCT116 cells in G1, S, and G2/M phases was determined using the Dean–Jett–Fox algorithm in (A). Statistical analysis was performed using an unpaired two-tailed Student's $t$ test (*$P < 0.05$) in (B). (C, D) Data are representative of $n = 3$ independent biological replicates. Each column represents relative expression values in independent biological replicates. Source data are available online for this figure.

phosphorylated ATM, ATR, CHK1, and CHK2, reaching levels observed in untreated acid-exposed cancer cells (pH$_e$ 6.5) (Fig. 4E). A similar pattern of ATM, ATR, and checkpoint kinase activation was observed in 5-FU-treated RER-negative HT-29 cancer cells at pH$_e$ 7.4 vs. untreated HT-29 cells at pH$_e$ 6.5 (Fig. 4F).

These findings led us to examine the sensitivity of both HCT116 and HT-29 cancer cells maintained at pH$_e$ 6.5 or 7.4 to selective inhibitors of ATM and ATR kinases, AZD0156 and KU60019 (ATMi) and elimusertib (ATRi), respectively. We found that ATMi significantly inhibited the growth of both HCT116 and HT-29 cancer cells cultured at pH$_e$ 6.5 but left unaltered both cancer cell types cultured at physiological pH$_e$ (Figs. 4G,H and EV4B,C); a small increase in cell growth was even observed at pH 7.4 in the presence of some ATMi concentrations (Figs. 4G,H and EV4B,C). ATRi also revealed preferential growth inhibitory effects in acid-exposed HCT116 and HT-29 cancer cells (Fig. 4I,J). We then examined whether DDRi could sensitize HCT116

and HT-29 cancer cells to the effects of 5-FU. We found that both acid-exposed cancer cells were more resistant to 5-FU (than corresponding cancer cells maintained at pH$_e$ 7.4) (see first two bars in Fig. EV4D–G). but exhibited an enhanced growth inhibitory response in the presence of ATRi (and ATMi for acid-exposed HT-29 cancer cells) (Fig. EV4D–G). Altogether, the above data suggest that ATMi and ATRi represent drugs of choice to target acid-exposed cancer cells as a single treatment but also in combination with chemotherapy.

## Combination of 5-FU with ATMi or ATRi inhibits the growth of 3D spheroids and large organoids derived from patient tumors

We next used 3D spheroids wherein both acidic and non-acidic compartments coexist to evaluate the resulting effects of combining chemotherapy with ATMi or ATRi. As a first insight on a potential

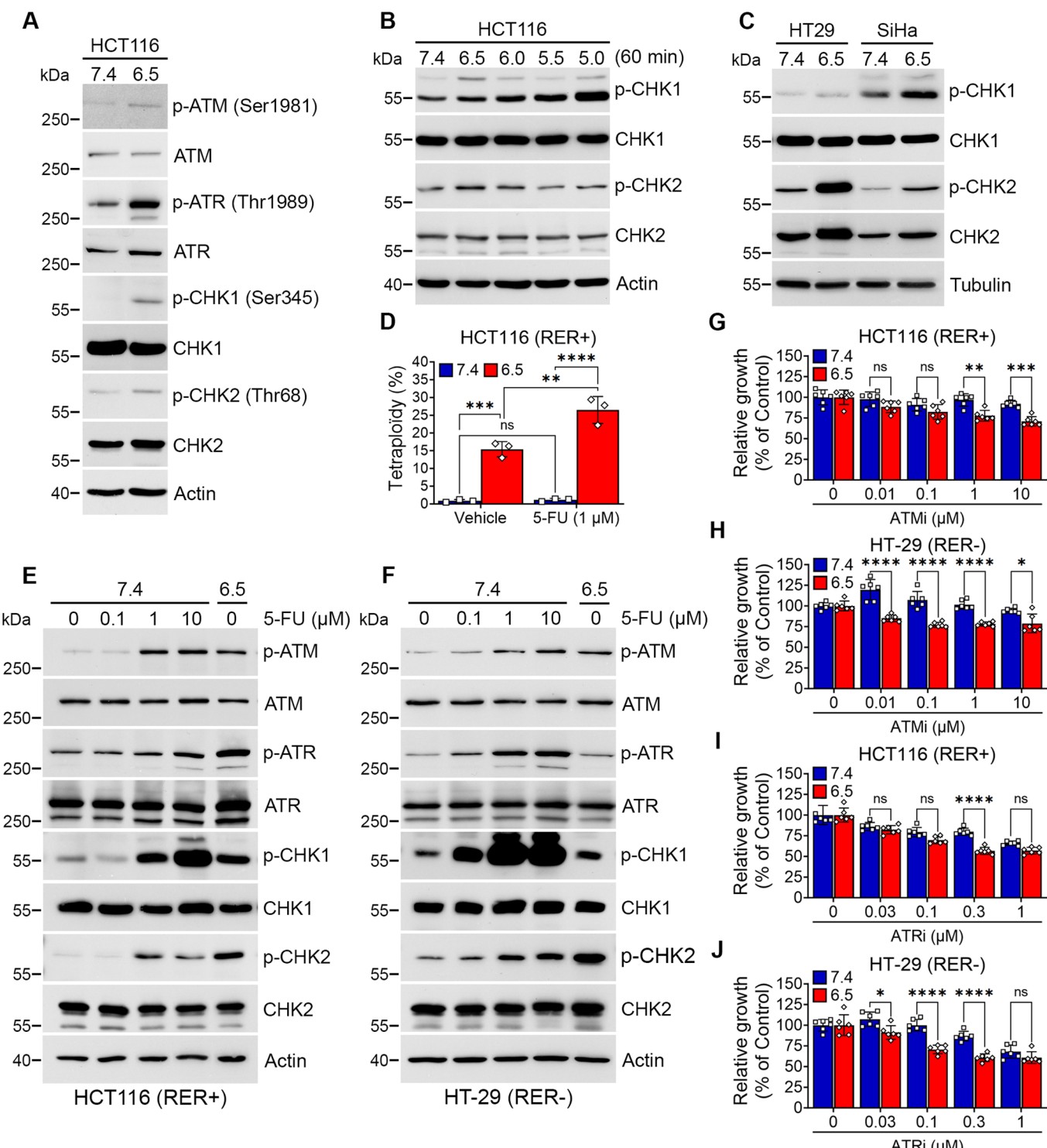

activation of these pathways in 3D spheroids, immunofluorescence revealed an increased phospho-ATM signal in the core but not the periphery of the spheroid, whereas the total ATM signal span to the whole spheroid (Fig. EV5A); phospho-ATR signal antibodies failed to reveal a specific signal in 3D spheroids in our hands. In HCT116 spheroids, ATMi and ATRi led to a significant growth inhibition when used as a single agent (see black bars in Fig. 5A,B) and enhanced the effects of 5-FU when concomitantly administered

(Fig. 5A,B). The observed inhibition in cell growth actually reflected both cytostatic and cytotoxic effects as revealed on pictures of treated spheroids (Appendix Fig. S1A–D). Of note, such beneficial effects of the combo (*vs.* single treatments) in p53 wild-type HCT116 spheroids were also obtained using spheroids made of p53-deficient HCT116 cells (Appendix Fig. S1E,F). Bliss synergy score calculation revealed that ATMi exerted additive effects when combined with 5-FU, whereas the combo 5-FU and ATRi were

Figure 4. Acid-exposed cancer cells activate ATM, ATR, and CHK1/2 kinases mimicking the response to chemotherapy.

HCT116, HT-29 and SiHa cancer cells were chronically adapted at pH 6.5 or maintained at pH 7.4 (A, C, D) or native HCT116 cells were acutely exposed to acidic pH$_e$ (B) or treated with 5-FU at the indicated doses to be compared with cancer cells adapted at pH 6.5 (E, F). (A–C) Representative immunoblots of total and phosphorylated ATM, ATR, CHK1, CHK2. Actin or tubulin was used as loading control, as indicated. (D) Bar graph showing the proportion of tetraploid cells. (E, F) Representative immunoblots of total and phosphorylated ATM, ATR, CHK1, CHK2. Actin was used as loading control, as indicated. (G–J) Cell viability assays in HCT116 (G, I) and HT-29 cancer cells (H, J) cultured at pH 7.4 or 6.5, and treated with the indicated doses of ATMi AZD0156 (G, H) or ATRi elimusertib (I, J) for 72 h. Data information: (A–J) data represent $n = 3$ independent biological replicates. (D, G–J) Bar graphs represent means ± SD with three biological replicates (D) or six technical replicates (G–J), and significance was determined using two-way ANOVA with Tukey's multiple-comparison analysis (ns non-significant; *$P < 0.05$; **$P < 0.01$; ***$P < 0.001$; ****$P < 0.0001$). Source data are available online for this figure.

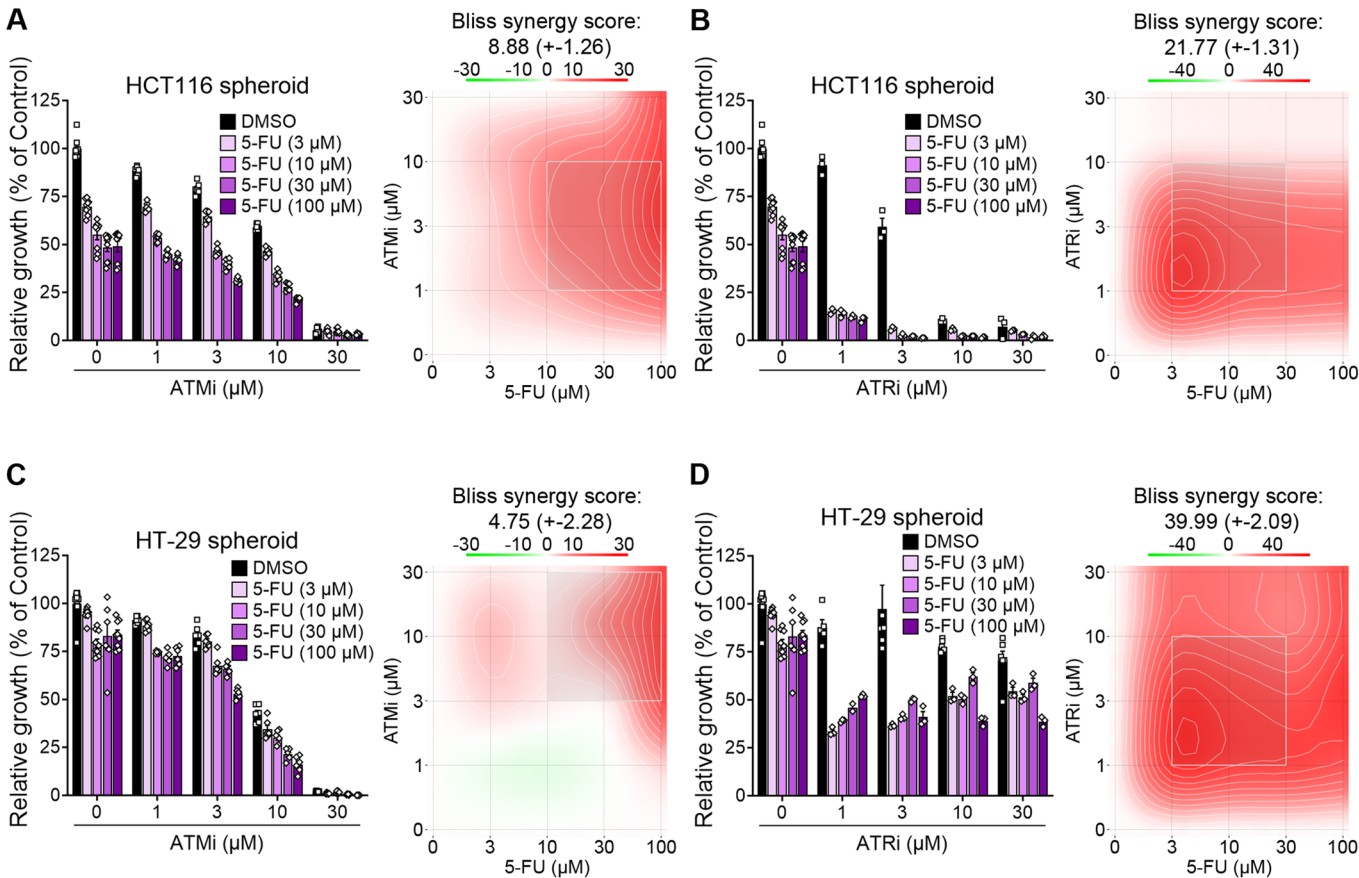

Figure 5. Growth inhibitory effects of a combination of 5-FU with ATMi or ATRi on 3D tumor spheroids.

(A–D) 3D cell viability assay in 3D HCT116 (A, B, left bar graphs) and HT-29 spheroids (C, D, left bar graphs) treated with the indicated doses of 5-FU combined with logarithmic dilutions of ATMi AZD0156 (A, C) or ATRi gartisertib (B, D) for 72 h. 2D synergy landscapes for serial dilutions of 5-FU and ATMi or ATRi are depicted (A–D, right panels). Data information: (A–D) bar graphs represent means ± SEM (3–12 technical replicates, $n = 3$ independent biological replicates). Statistical evaluation of drug combination effect was calculated using the Bliss independence dose-response model. Bliss synergy scores (± SD) and color legend are indicated. Source data are available online for this figure.

synergistic (see right panels in Fig. 5A,B). In HT-29 spheroids, higher intrinsic 5-FU resistance was observed (see first five left bars in Fig. 5C) but again combo treatment revealed additive effects from 3 µM ATMi and synergistic effects from the smallest ATRi concentration (Fig. 5C,D). Finally, we used colorectal cancer patient-derived tumor organoids (PDTO) that we treated with single drugs (ATMi or 5-FU) or combo. Using large organoids to maximize the development of acidic compartments, we found that their growth was significantly inhibited by the combo drugs (*vs.* either single drug) (Fig. EV5C,D). Altogether, these data indicate that tumor acidosis

represents a prognostic marker of CRC progression but also a potential predictive biomarker of the efficacy of ATMi and ATRi used alone or in combination with chemotherapy.

## Discussion

The main finding of this study is the identification of tetraploidy and DDR as major hallmarks but also a druggable vulnerabilities of acid-exposed cancer cells, a population of cells actively participating to

tumor invasiveness and anticancer drug resistance (Corbet et al, 2020; Corbet and Feron, 2017; Ibrahim-Hashim and Estrella, 2019; Pillai et al, 2019; Rohani et al, 2019; Wojtkowiak et al, 2011; Yao et al, 2020). Importantly, this characteristic was unraveled from cancer cells isolated from 3D tumor spheroids wherein acidosis spontaneously develops, thereby adding an important stamp of relevance when compared with cancer cell monolayers. The latter indeed do not grow in an environment that integrates the complexity of in vivo tumor conditions as recapitulated in 3D spheroids, including the heterogeneity of cancer cells exposed to pH, $pO_2$, and nutrient/waste gradients. Acidosis, as well as hypoxia, have often been considered as overlapped characteristics of tumors, mostly because of the release of high amounts of $H^+$ by highly glycolytic hypoxic cancer cells. We and other have contributed to show that acidosis covers a larger proportion of cancer cells within a tumor, including a cell population being more dependent on fatty acid oxidation and OXPHOS, two $O_2$-dependent metabolic pathways (Corbet et al, 2020; Corbet et al, 2016; Lan et al, 2022; Michl et al, 2022; Rolver et al, 2023). The pHLIP-based sorting approach used in this study confirmed the enrichment of these two pathways in acid-exposed cancer cells. While these observations validated our methodology, the current work also highlighted a robust DDR in acid-exposed cancer cells together with an accumulation of tetraploid cells, supporting the occurrence of a strong genotoxic stress under acidosis.

Tumor acidosis can contribute to accumulation of DNA damages through various mechanisms. Among them, our pathway enrichment analysis supports a role for increased production of reactive oxygen species which can cause oxidative damage to DNA (Gorrini et al, 2013; Maldonado et al, 2023) as well as stimulated histone deacetylation and associated alterations in the regulation of DNA repair processes (Roos and Krumm, 2016). Indeed, histone deacetylation is known to promote both a condensed chromatin structure which restricts the accessibility of DNA repair machinery to damaged sites, and alteration in the stability of chromatin making it more prone to DNA insults (Aricthota et al, 2022; Robert and Rassool, 2012; Roos and Krumm, 2016; Thurn et al, 2013). Interestingly, we previously documented that histone deacetylation was stimulated in acid-exposed cancer cells to provide them with acetate as a counter-anion to export excess cytosolic $H^+$ through MCT carriers (Corbet et al, 2016). We originally identified high cytosolic $NAD^+$ levels associated with enhanced fatty acid oxidation and reduced glucose metabolism as the driver of enhanced SIRT1 and SIRT6 activities. Our gene enrichment data analysis now allows to associate a broad HDAC cluster in acid-exposed cancer cells with an increased incidence in DNA lesions and DDR activation.

The significant activation of checkpoint proteins in acid-exposed cancer cells indicates that repair pathways are induced, minimizing the transmission of damaged DNA to daughter cells. However, even if growth of acid-exposed cancer cells is maintained, a significant increase in tetraploid cells (i.e., 4 N DNA content) was observed, indicating that genomic instability affects a significant proportion of acid-exposed cancer cells. Whole genome doubling cancer cells can result from various mechanisms including cell fusion, endoreplication, mitotic slippage and cytokinesis failure (Vittoria et al, 2023). While cell fusion is most often associated with viral infection (Duelli et al, 2007), drivers of other pathways may be manifold and intertwined. For instance, failure of cell division may promote endoreplication and inversely, leading to a further augmentation of DNA content (Shu et al, 2018). Together with our observations that acid-exposed cancer cells (from different origins) are slightly bigger than corresponding cancer cells cultured at

physiological $pH_e$, our findings are in line with a previous study reporting endoreplication in response to lactic acidosis (Tan et al, 2018). On the long term, since tetraploid cells are more prone to additional errors during subsequent divisions, chromosomal abnormalities of acid-exposed cancer cells are very likely to contribute to genetic instability and heterogeneity favorable to tumor progression and drug resistance, respectively. Whether these tetraploid cells are related to the so-called polyploid giant cells (Illidge et al, 2000; Shu et al, 2018) warrants further investigation. The latter indeed share key hallmarks with acid-exposed cancer cells, namely a role in tumor recurrence and escape from DNA-damaging drugs (Puig et al, 2008; Zhou et al, 2022). Also, of potential therapeutic interest, our gene enrichment study (Fig. 2A,B) identified in acid-exposed cancer cells an increase in the signaling pathway driven by POLO-like 1 kinase (PLK-1) that is known to support mitotic entry following recovery from DNA damage in polyploid cells (van Vugt et al, 2004). Our preliminary exploration of BI-2536, a PLK-1 inhibitor, revealed that while the growth inhibitory effects were similar in CRC cells kept at pH 7.4 compared to those exposed to pH 6.5 (even less pronounced for the latter), the reappearance of cell growth following drug removal was notably reduced in the presence of acidosis (Appendix Fig. S2). These findings support previous observations of polyploid cells being more readily moved toward mitotic catastrophe-induced apoptosis upon PLK-1 inhibition or silencing (Jemaa et al, 2020). Further experiments are warranted to evaluate the safety of such approaches and determine whether other therapeutic strategies targeting tetraploid cells may be more appropriate (Coward and Harding, 2014; Quinton et al, 2021).

Interestingly, we found that acid-exposed cancer cells exhibited a preferred response to inhibitors of ATM and ATR (vs. corresponding cells at physiological $pH_e$). Remarkably, when evaluating effects of these drugs on 3D spheroids and patient-derived tumor organoids wherein both acidic and non-acidic cancer cells coexist, combination with 5-FU led to additive and even synergistic effects. In particular, a low dose ATRi which per se did not induce significant growth inhibitory effects, led to a dramatic increase in the cytotoxic effects of 5-FU in either HCT116 or HT-29 spheroids. It is worth to note that while HCT116 cells are DNA mismatch repair-deficient (Bracht et al, 2010; Kennedy et al, 2000), our data indicate that acidosis may still act as a revelator of the therapeutic potential of ATMi or ATRi. Also, while RER-negative HT-29 CRC cells harbor a similar increased capacity to respond to DNA lesions when facing an extracellular acidic environment, they are as resistant to 5-FU if not more so than HCT116 CRC cells (Varghese et al, 2019) (see also Fig. 5). The emergence of HT-29 cell resistance to 5-FU is actually thought to result from the progressive selection for a cancer stem cell (CSC) subpopulation (Dallas et al, 2009). Altogether, while deficits in DNA mismatch repair systems and acquisition of CSC phenotype certainly participate to determine the sensitivity of cancer cells to genotoxic drugs (Bracht et al, 2010; Touil et al, 2014), our data document that microenvironmental acidosis increases the basal DDR extent and may at the very least exacerbate it in response to chemotherapy, thereby increasing the interest of a combo treatment to target the aggressive acidic tumor compartment.

As it would be in mouse tumors, is difficult to address the spatio-temporal pharmacodynamics of increasing doses of 5-FU and either ATMi or ATRi in 3D spheroids (i.e., the kinetics and extent of cell death occurring in the acidic vs. non-acidic

compartments). Still, our study identified tumor acidosis as a new unequivocal player in the rationale of combining genotoxic chemotherapy with drugs disrupting DNA repair. As pointed out by Nickoloff et al, the therapeutic index of DDRi in many cancers may not necessarily result from a differential repair capacity between normal and cancer cells, but rather from the heavier load of DNA damage (Nickoloff, 2022; Nickoloff et al, 2017). Acidosis-induced DDR may actually reflect such propensity of a proportion of cancer cells within many tumors to face mutational stress conditions, thereby also suggesting that DNA repair targeting strategies could be more widely applied.

In conclusion, our study presents evidence that induction of DDR is a hallmark of any cancer cells experiencing an acidic milieu and that ATM and ATR inhibitors can directly take advantage of this context. This finding has an immediate potential for transfer research at a time when clinical trials are being conducted to evaluate the safety, efficacy, and optimal combinations of ATMi or ATRi with different chemotherapy drugs for specific cancer types. Identifying reliable biomarkers to select patients who are most likely to benefit from ATM or ATR inhibitors is actually a major challenge. The current study strongly suggests that evaluating the extent of tumor acidosis could represent an attractive approach to predict responsiveness to these inhibitors, allowing for more personalized treatment approaches. Implementation of this patient stratification strategy could take advantage of recent developments in CEST-MRI, a noninvasive technology offering high spatial resolution and sensitivity for in vivo imaging of tumor acidosis (Anemone et al, 2021; Jardim-Perassi et al, 2023), and would certainly gain in being combined to composite biomarkers recently set up to identify patients more likely to respond to DDRi (Durinikova et al, 2022).

# Methods

## Reagents and tools table

See Table 1.

**Table 1.  Reagents and tools.**

| Reagent/resource | Reference or source | Identifier or catalog number |
|---|---|---|
| **Experimental models** | | |
| HCT116 cells (*H. sapiens*) | ATCC | CCL-247; RRID:CVCL_0291 |
| HT-29 cells (*H. sapiens*) | ATCC | HTB-38; RRID:CVCL_0320 |
| SiHa cells (*H. sapiens*) | ATCC | HTB-35; RRID:CVCL_0032 |
| HCT116 p53-deficient cells (*H. sapiens*) | kind gift from Chris Marine, Laboratory for Molecular Cancer Biology, Center for Cancer Biology, VIB, Louvain, Belgium | |
| Patient-derived Organoids (*H. sapiens*) | Local library of organoids collected from CRC patients undergoing surgery or tumor biopsy at Cliniques Universitaires St Luc, Brussels | |
| **Antibodies** | | |
| Mouse monoclonal anti-β-actin (clone AC-15) | Sigma-Aldrich | Cat# A5441; RRID: AB_476744 |
| Rabbit monoclonal anti-phospho-ATM (Ser-1981) (clone D25E5) | Cell Signaling Technology | Cat# 13050; RRID: AB_2798100 |
| Rabbit monoclonal anti-ATM (clone D2E2) | Cell Signaling Technology | Cat# 2873; RRID: AB_2062659 |
| Rabbit monoclonal anti-phospho-ATR (Thr-1989) (clone D5K8W) | Cell Signaling Technology | Cat# 30632; RRID: AB_2798992 |
| Rabbit monoclonal anti-ATR (clone E1S3S) | Cell Signaling Technology | Cat# 13934; RRID: AB_2798347 |
| Rabbit monoclonal anti-phospho-CHK1 (Ser-345) (clone 133D3) | Cell Signaling Technology | Cat# 2348; RRID: AB_331212 |
| Mouse monoclonal anti-CHK1 (clone 2G1D5) | Cell Signaling Technology | Cat# 2360; RRID: AB_2080320 |
| Rabbit monoclonal anti-phospho-CHK2 (Thr-68) (clone C13C1) | Cell Signaling Technology | Cat# 2197; RRID: AB_2080501 |
| Mouse monoclonal anti-CHK2 (clone 1C12) | Cell Signaling Technology | Cat# 3440; RRID: AB_2229490 |
| Rabbit polyclonal anti-α/β-tubulin | Cell Signaling Technology | Cat# 2148; RRID: AB_2288042 |
| **Chemicals, enzymes, and other reagents** | | |
| AZD0156 | MedChemExpress | Cat# HY-100016 |
| Elimusertib | MedChemExpress | Cat# HY-101566 |

**Table 1.** (continued)

| Reagent/resource | Reference or source | Identifier or catalog number |
|---|---|---|
| Gartisertib | MedChemExpress | Cat# HY-136270 |
| KU60019 | MedChemExpress | Cat# HY-12061 |
| 5′Fluorouracil | N/A | N/A |
| DAPI | Sigma-Aldrich | Cat# D9542 |
| Faramount Aqueous Mounting Medium | Dako | Cat# S3025 |
| Neg-50 Frozen Section Medium | Thermo Fisher Scientific | Cat# 6502Y |
| Cultrex Basement Membrane Extract, Type 2 | Thermo Fisher Scientific | Cat# 3532-005-02 |
| Propidium iodide | Sigma-Aldrich | Cat# P4170 |
| Dulbecco's Modified Eagle's Medium (DMEM), High glucose, GLUTAMAX supplement | Thermo Fisher Scientific | Cat# 61965-026 |
| Penicillin/streptomycin | Gibco | Cat# 15140-122 |
| Fetal bovine serum (FBS) | Sigma-Aldrich | Cat# F6765 |
| PhosSTOP | Roche | Cat# 4906845001 |
| Protease Inhibitor Cocktail | Sigma-Aldrich | Cat# P8340 |
| RNase A | Thermo Fisher Scientific | Cat# EN0531 |
| Triton X-100 | Thermo Fisher Scientific | Cat# HFH10 |
| Trypan Blue | Thermo Fisher Scientific | Cat# 15250061 |
| pH-(low)-insertion peptide (pHLIP) v3 conjugated to Alexa Fluor 568 | (Weerakkody et al, 2013) | N/A |
| K-pHLIP conjugated to Alexa Fluor 594 | (Weerakkody et al, 2013) | N/A |
| **Software** | | |
| FlowJo software v10 | https://www.flowjo.com/ | N/A |
| GraphPad Prism 10 | https://www.graphpad.com | N/A |
| ImageJ (Fiji) | https://imagej.net/ij/ij/index.html https://fiji.sc/ | N/A |
| Gene Set Enrichment Analysis (GSEA) | https://www.gsea-msigdb.org/gsea/index.jsp | N/A |
| Metascape | https://metascape.org/gp/index.html#/main/step1 (Zhou et al, 2019) | N/A |
| g:Profiler | https://biit.cs.ut.ee/gprofiler/gost (Raudvere et al, 2019) | N/A |
| iDEP.96 | http://bioinformatics.sdstate.edu/idep96/ (Ge et al, 2018) | N/A |
| SynergyFinder 3.0 (Bliss score) | https://synergyfinder.fimm.fi/ (Ianevski et al, 2022) | N/A |
| **Other** | | |
| Bicinchoninic acid-based protein assay kit | Thermo Fisher Scientific | Cat# 23225 |
| CellTiter-Glo® 3D Cell Viability assay | Promega | Cat# G9682 |
| PrestoBlue™ Cell Viability Reagent | Thermo Fisher Scientific | Cat# A13261 |
| TRIzol Reagent | Thermo Fisher Scientific | Cat# 15596026 |

## 2D cell cultures

HCT116, HT-29, SiHa cancer cell lines were purchased from the American Type Culture Collection (ATCC). All cells were stored according to the supplier's recommendations and used within 20 passages after resuscitation of frozen aliquots. All cells were regularly tested with the Mycoplasma Detection kit (Lonza) to exclude mycoplasma contamination. Cell lines were cultured in Dulbecco's Modified Eagle Medium (DMEM) (GlutaMAX, 25 mM D-glucose; Gibco 61965-026) supplemented with 10% heat-inactivated Fetal Bovine Serum (FBS) (Sigma F6765) and 1% penicillin/streptomycin (P/S) (Gibco 15140-122), in standard humidified 5% CO2/37 °C incubators. HCT116, HT-29 and SiHa cell lines were also adapted to an acidic medium to reflect the conditions observed in some areas of solid tumors as previously documented (Corbet et al, 2020; Corbet et al, 2014; Corbet et al, 2016). To ensure the pH stability in such a medium, cells were maintained in a standard medium consisting of DMEM (Sigma-Aldrich) supplemented with 10% heat-inactivated FBS, 1% P/S, 10 mM D-glucose, 2 mM L-glutamine, 25 mM HEPES and PIPES, and adjusted to (control) physiological pH 7.4 or to pH 6.5. Prior to filter sterilization, this standard medium was split, adjusted to pH 7.4 or pH 6.5, and brought to volume, allowing identical media composition of physiological and acidic media in all other regards.

## 3D tumor spheroids

HCT116 and HT-29 spheroids were initiated by seeding cell lines at a given density (750 cells/well for HCT116; 1000 cells/well for HT-29) in ultra-low attachment 96-well plates (Corning, costar 7007), in DMEM supplemented with 10% heat-inactivated FBS and 1% P/S. Cell number was assessed by cell counting on a Cellometer Auto T4 Bright Field Cell Counter (Nexcelcom) with Trypan blue (Gibco 15250061) exclusion dye. After seeding and a centrifugation step (5 min; 600 rpm), one spheroid per well spontaneously formed after three days of culture in a conventional humidified 5% $CO_2$/37 °C incubator. Spheroid growth was monitored using live-cell phase contrast microscope (AxioObserver, Zeiss). In order to work with spheroids with minimal necrotic areas but large enough to let $pO_2$ and pH gradients to develop, experiments were performed within a period of 4–7 days.

For immunofluorescence studies, spheroids were washed twice in PBS, fixed in 4% PFA, harvested, and embedded in OCT matrix (Thermo Fisher Scientific). Frozen sections (5 μm) were used for detection of Alexa-conjugated pHLIP and K-pHLIP peptides and nuclei were counterstained with DAPI. Alternatively, spheroids were washed twice in PBS, fixed in 4% PFA, harvested, incubated in 20% sucrose, and embedded in 7.5% gelatin containing 15% sucrose. Frozen sections (8 μm) were used for detection of phospho-ATM, ATM, and nuclei were counterstained with DAPI. Slides were prepared with fluorescence mounting medium (Dako), and staining was visualized with a Zeiss Imager 1.0 Apotome microscope. All spheroid samples from a same experiment were imaged by using the same gain and exposure settings.

## Patient-derived tumor organoids

Tumor organoids were generated from local library of specimens collected from CRC patients undergoing surgery or tumor biopsy at Cliniques Universitaires St Luc in Brussels (ethics committee approval no. ONCO-2015-02 updated on 13-05-2019 in accordance with the principles of the Declaration of Helsinki). Signed informed consent was obtained from all the patients before tumor collection and all personal/clinical data remained strictly confidential for the investigators. Organoids were processed to single cells and seeded with a density of 40,000 cells per well in 24 multiwell plates (Corning™ Primaria™ Cell Culture Multiwell™ Plates, Corning™ 353847), embedded in Cultrex matrix (Cultrex Basement Membrane Extract, Type 2, Thermo Fisher Scientific 3532-005-02). After 1 week, organoids acquired their typical 3D conformation and were further expanded for 7 days before treatment for 96 h with 1 μM AZD0156 and/or 10 μM 5-FU. Organoid growth was monitored in time-lapse with IncuCyte SX5® Live-Cell Analysis system (Essen Bioscience).

## FACS-based isolation of acid-exposed cancer cells

3D tumor spheroids were incubated for 24 h with 2 μM Alexa 568-conjugated pH-low insertion peptide variant 3 (pHLIP V3; NH₂-ACDDQNPWRAYLDLLFPTDTLLLDLLW-COOH) (Weerakkody et al, 2013) to label acidic regions; an acid-independent Alexa 594 K-pHLIP peptide wherein aspartate residues were replaced by positively charged lysine residues was used as negative control. After cell dissociation using Accutase, cells were resuspended into PBS for FACS sorting. FACSAria III cell sorter (BD Biosciences) was used to isolate Alexa 568 pHLIP-positive and Alexa 594 K-pHLIP-negative cells.

## Cell cycle analysis

Asynchronous cells were grown in medium supplemented with 10% FBS for 24 h prior to changing media change. The DNA and RNA intercalating fluorescent dye propidium iodide (PI, Sigma-Aldrich, P4170) was used to quantify cellular DNA content and cell cycle distribution. After treatment, cells were harvested, resuspended in PBS containing 2% FBS, washed once in PBS, and fixed in ice-cold 95% (v/v) ethanol for at least 24 h at −20 °C. The cell pellet was collected by centrifugation at 500×g, gently resuspended in PBS, washed in PBS and stained with a mixture of RNase A (100 μg/mL; Thermo Fisher Scientific, EN0531) and the PI (40 μg/mL) in PBS for 30 min in the dark. Samples were immediately analyzed by flow cytometry with a FACSCalibur flow cytometer (BD Biosciences). A total of 20,000 events was recorded per sample, and the cell cycle fraction G1, S, and G2/M phases were quantified using the Dean–Jett-Fox or Watson algorithm of FlowJo software v10.

## SDS-PAGE and immunoblotting (IB)

Cells were washed twice with cold PBS 1× (pH 7.4) on ice and lysed in cell RIPA lysis buffer (50 mM Tris-HCl pH 7.4, 1 mM EDTA, 150 mM NaCl, 1% Triton X-100, 0.1% SDS, 0.05% Sodium deoxycholate, Phosphatase Inhibitor Cocktail (Sigma-Aldrich, P8340), and PhosSTOP (Roche, 4906837001)) for 30 min at 4 °C. After collection, the total cell lysates were centrifuged (10 min at 10,000 rpm; 4 °C) to recover the protein supernatant. Protein samples were quantified with the Bicinchoninic acid (BCA) method (Thermo Fisher, 23225) using SpectraMaxI3 instrument (562 nm). All protein samples were then prepared at a concentration of 2 μg/μL in a 6× Laemmli buffer (375 mM Tris-HCl pH 6.8, 30% glycerol, 10% SDS, 60 mM DTT, and 0.03% bromophenol blue) and heated for 7 min at 95 °C. For IB analysis, protein samples were subjected to 8–10% SDS-PAGE, and resolved proteins were transferred onto polyvinylidene fluoride (PVDF) membrane (PerkinElmer, NEF1002001PK). Membranes were then blocked for 1 h in a TBS-Tween solution (10 mM Tris pH 7.4, 0.1% Tween 20) supplemented with 5% (w/v) dry skim milk powder, and subsequently immunoblotted with primary antibodies of interest, diluted in 5% (w/v) dry skim milk or 5% (w/v) bovine serum albumin (BSA), overnight at 4 °C. Antibodies targeted against phosphorylated p-ATM (D25E5) (Ser-1981) (1:1000), ATM (D2E2) (1:1000), p-ATR (D5K8W) (Thr-1989) (1:1000), ATR (E1S3S) (1:1000), p-CHK1 (133D3) (Ser-345) (1:1000), CHK1 (2G1D5) (1:1000), p-CHK2 (C13C1) (Thr-68) (1:1000), CHK2 (1C12) (1:1000) and α/β-tubulin (1:2000) are from Cell Signaling Technologies; β-Actin (AC-15) (1:30,000) from Sigma-Aldrich. After several washes with TBS-Tween, membranes were incubated with 1 h at room temperature with the secondary horseradish peroxidase (HRP)-conjugated antibodies (Jackson Immunoresearch), diluted in TBS-Tween 1% milk (1/5,000), and chemiluminescent signals were revealed using ECL Western Blotting Detection kit (Amersham ECL, RPN2134) on photographic films (Thermo Fisher Scientific, 34089) in a dark room. All IB data are representative of at least three independent biological experiments.

## Cell viability assays

For 2D cell cultures, cells were grown in medium supplemented with 10% FBS, and relative number of viable cells was assessed with the PrestoBlue reagent (ThemoFisher Scientific, A13262). In all, 10× diluted PrestoBlue was added in the dark 1 h prior to fluorescence measurement (Ex/Em=560/590 nm) using the SpectramaxI3 microplate reader (Molecular Devices, LLC). Cells were plated and after adhesion were treated as a monotreatment with ATMi (AZD0156 [HY-100016], KU60019 [HY-12061], MedChemExpress) or ATRi (Gartisertib [HY-136270], Elimusertib [HY-101566], MedChemExpress) at the indicated concentration. Cells were grown for 72 h before addition of PrestoBlue. Data displayed correspond to the mean of six technical replicates ± standard deviation (SD) and are representative of $n = 3$ independent experiments. Data were normalized as the relative percentage of the signal detected in the absence of drug.

For 3D cell cultures, spheroids were grown in ultra-low-attachment 96-well plates for 7 days, and viability was assessed using the CellTiter-Glo luminescent viability assay kit (Promega, G9682). 4 days after initiation, spheroids were treated with 5-FU as a monotherapy or in combination with AZD0156 or Gartisertib for 72 h. The spheroids were then transferred to an opaque 96-well plate (Greiner Bio-One, Lumitrac) in a volume of 100 μL and mixed with 1:1 volume of CellTiter-Glo reagent under orbital shaking to induce cell lysis (5 min; 900 rpm). After 30 min of incubation in the dark at room temperature, luminescence was read at 560 nm with the SpectramaxI3. Data displayed correspond to the mean of six individual spheroids ± standard error of the mean (SEM) and are representative of $n = 3$ independent experiments. Data were normalized as the relative percentage of the signal detected in the absence of drug. Bliss scores were calculated using the online tool SynergyFinder 3.0 (Ianevski et al, 2022).

## RNA-sequencing and transcriptome analyses

The transcriptome of negative K-pHLIP- and positive pHLIP-sorted cells from HCT116 spheroids were analyzed by RNA-seq. Total RNA was isolated from cells using TRIzol reagent, according to the manufacturer's instructions. rRNA-depleted RNA was used for the library preparation with the NEBNext Ultra II Directional RNA library prep kit for Illumina (New England Biolabs) according to the manufacturer's instructions. Libraries were paired-end sequenced (2*76 base pairs) on a NextSeq500 device (Illumina), with a read depth of 20 million. Kallisto software (Bray et al, 2016) was used for quantifying transcript abundance from RNA-seq data against GRCh38 cDNA reference transcriptome from the Ensembl database, v96. Expression levels of mRNA were displayed as transcripts per million (TPM); these TPM values were used to analyze the functional enrichment of genes associated with the sequencing data by Gene Set Enrichment Analysis (GSEA) against the 50 cancer hallmark gene sets from the MSigDB database (https://www.gsea-msigdb.org/gsea/msigdb/). Data are representative of three independent biological experiments. Principal Components Analysis (PCA) was performed to verify the consistency of biological replicates. TPM values of each biological replicate were averaged for non-acidic and acidic conditions. Only transcripts of protein-coding genes with an expression cutoff of Log$_2$ FC (acid/non-acid) >1 or Log$_2$ FC (acid/non-acid) <1 and a significance threshold of $P < 0.05$, were respectively considered upregulated or downregulated at low pH. Two separate functional enrichment analyses were conducted either on upregulated protein-coding genes using the web-based tool g:Profiler (version e109_eg56_p17_1d3191d) (Raudvere et al, 2019) against Gene Ontology Cellular Component (GO:CC) and Reactome (Gillespie et al, 2022) pathway databases, or on both upregulated and downregulated protein-coding genes via the online platform Metascape (Zhou et al, 2019). Alternatively, differential gene expression analysis was also carried out using DESeq2 package via the web-based tool iDEP.96 (Ge et al, 2018), and parametric GSEA were conducted using the iDEP workflow against the Wikipathways (WP) (Martens et al, 2021) and Pathway Interaction Database (PID) (Schaefer et al, 2009) pathway databases. The statistical significance threshold for all functional annotation overrepresentations was set on adjusted $P$ value (also referred to as $q$ value) <0.05. Pathway and gene interaction networks were generated using Metascape. The heatmaps in Figs. 3C,D and EV2F were generated by calculating, for each biological replicate, the z-score relative expression of genes encompassing the distinct functional terms ATM signaling (WP2516), ATR signaling (WP4016) and DNA repair (GO0006281). Only expressed genes with Log$_2$ FC (acid/non-acid) > |0.5| and $P$ value < 0.05 were displayed.

## Quantification and statistical analysis

The experiments were all performed in six technical replicates and repeated in $n \geq 3$ independent experiments unless specified. All analyses were performed blindly. For all relevant panels unless specified, center values and error bars represent means ± SD or SEM, and $P$ values were determined by unpaired two-tailed Student's $t$ tests used for comparisons between two groups, otherwise using two-way ANOVA with post hoc Tukey's multiple-comparison analysis. ns not significant; $*P < 0.05$, $**P < 0.01$; $***P < 0.001$; $****P < 0.0001$. Statistical measurements were made on distinct samples using both GraphPad PRISM v10 and Microsoft Excel softwares. Bliss synergy scores were carried with the SynergyFinder 3.0 platform. The interaction between two drugs is likely to be *synergistic* if the score is >10, *additive* if the score is between −10 and 10 and *antagonistic* if the score is < −10.

# Data availability

The dataset from RNA-sequencing analysis, generated during this study, is available at Gene Expression Omnibus GSE238108.

# Peer review information

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

## Acknowledgements

This work was supported by grants from the Fonds de la Recherche Scientifique (F.R.S.-FNRS, PDR T008719F and EOS O002522F), WELBIO (Walloon Excellence in Life Sciences and Biotechnology, X104022F), the (Belgian) Foundation against cancer (2020-074), the J. Maisin Foundation (2020-21) and an Action de Recherche Concertée (ARC 19/24-096). CC is a F.R.S.-FNRS Research Associate and OF is a Walloon Excellence in Life Sciences and Biotechnology (WELBIO) senior fellow.

## Author contributions

**Léo Aubert**: Conceptualization; Data curation; Formal analysis; Supervision; Validation; Investigation; Visualization; Methodology; Writing—original draft; Writing—review and editing. **Estelle Bastien**: Formal analysis; Investigation; Methodology. **Ophélie Renoult**: Formal analysis; Investigation. **Céline Guilbaud**: Formal analysis; Validation; Investigation. **Kubra Ozkan**: Formal analysis; Investigation; Methodology. **Davide Brusa**: Formal analysis; Methodology. **Caroline Bouzin**: Formal analysis; Methodology. **Elena Richiardone**: Methodology. **Corentin Richard**: Methodology. **Romain Boidot**: Methodology. **Daniel Léonard**: Methodology. **Cyril Corbet**: Methodology. **Olivier Feron**: Conceptualization; Resources; Supervision; Funding acquisition; Validation; Investigation; Visualization; Writing—original draft; Project administration; Writing—review and editing.

## Disclosure and competing interests statement

The authors declare no competing interests.

# Expanded View Figures

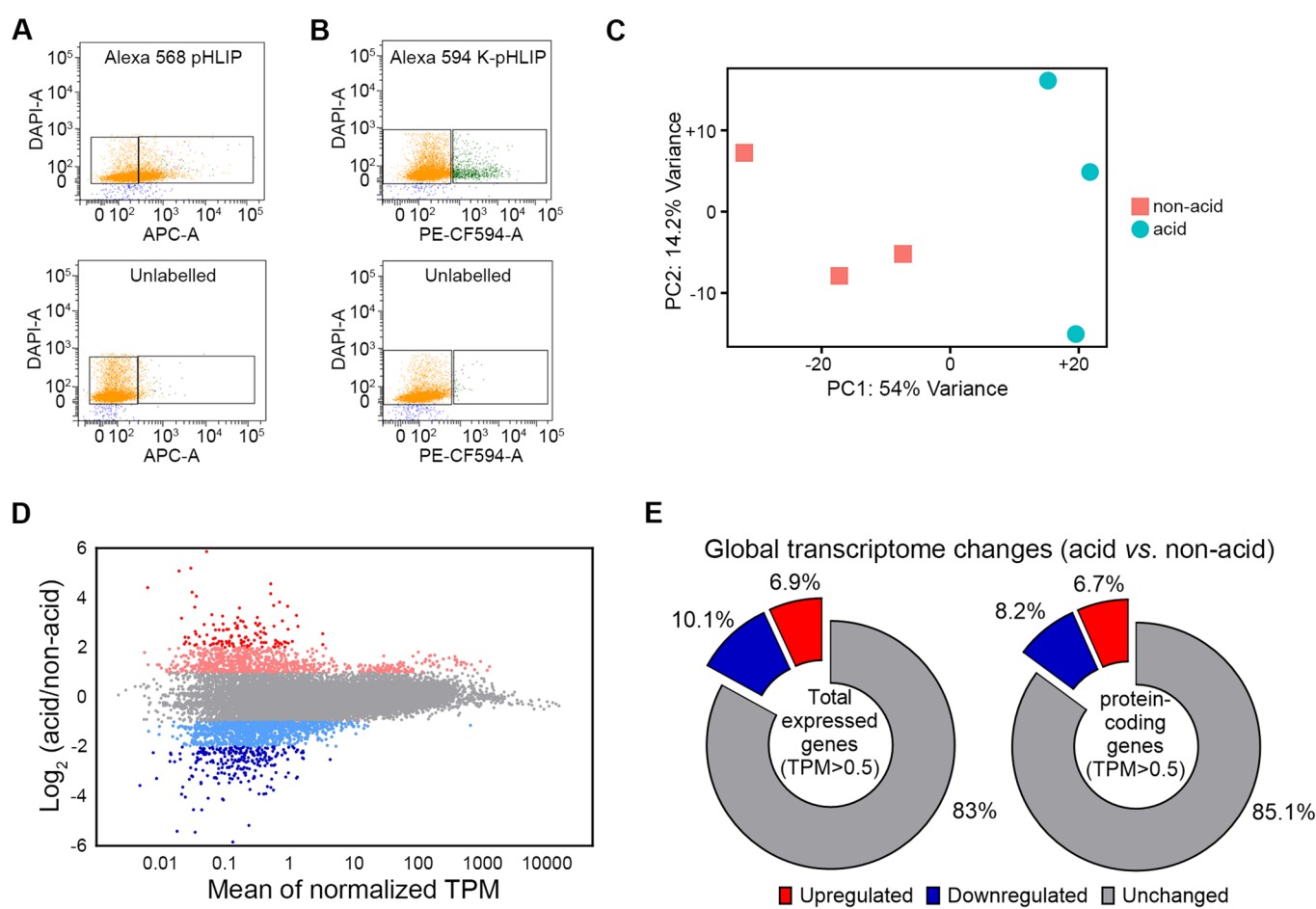

**Figure EV1.  Sorting of pHLIP-positive and K-pHLIP-positive cancer cells from 3D tumor spheroids and visualization of global transcriptome changes.**

(A, B) Dot plots of DAPI signal *vs.* Alexa 568 pHLIP (A) and Alexa 594 K-pHLIP (B) fluorescence intensities in cells sorted from 3D HCT116 spheroid. (C) PCA plot for the acid *vs.* non-acid-independent biological replicates ($n = 3$) of the 15,600 identified genes (TPM > 0.5). (D) MA-plot of the global transcriptome changes associated with acidic $pH_e$ value. Upregulated (red) or downregulated (blue) genes are shown (Log$_2$ FC (acid/non-acid) > |1| ). (E) Donut-plots showing the proportion of total expressed genes (left side) and protein-coding genes (right side) upregulated (red) or downregulated (blue) in acid-exposed cancers cells (TPM > 0.5; *P* value ≤ 0.05). Data information: Data are representative of $n = 3$ independent biological replicates. (E) *P* values filtering was determined from statistics in Fig. 1E.

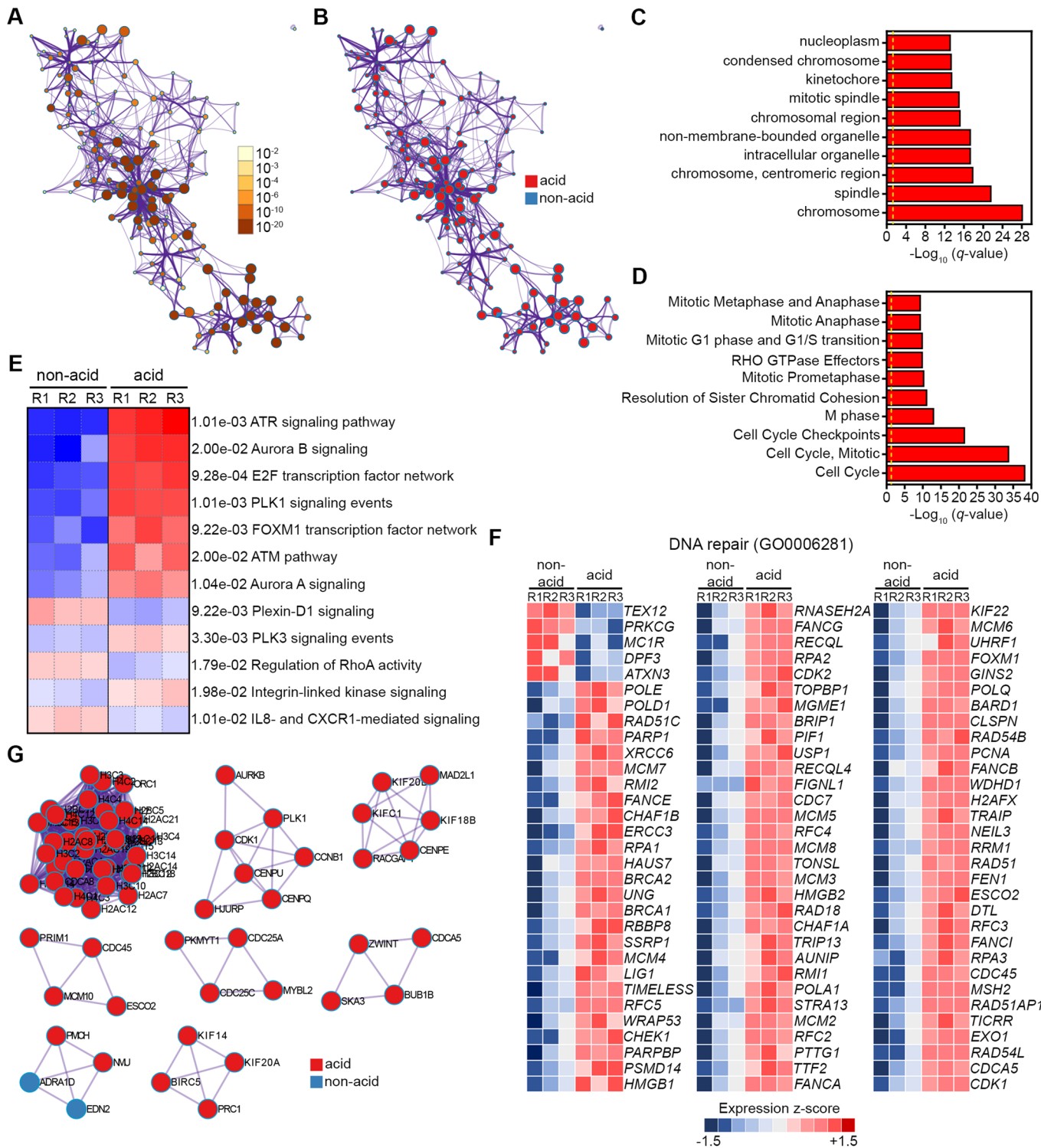

◀ **Figure EV2.   DDR signature in acid-exposed cancer cells isolated from 3D tumor spheroids.**

(A, B) Network analysis of Metascape-annotated functional clusters for (top-20) DE protein-coding genes upregulated or downregulated in acid-exposed cancer cells isolated from 3D spheroids. Each circle node represents a distinct pathway annotation. The thickness of the purple edges indicates the number of common genes between various pathway annotations. (A) Significance is indicated by the darkness of the node's color while in (B), red (upregulated) or blue (downregulated) color indicates the number of enriched genes in that pathway annotation. (C, D) Bar charts depicting the most significant Gene Ontology (GO): Cellular Component (CC) terms (C) and Reactome pathway (D) enrichment correlating with transcripts upregulated in acid-exposed cancer cells ($Q < 0.05$). (E) PGSEA analysis using Pathway Interaction Database (PID) revealed that many DDR-related gene sets are positively enriched in acid-exposed cancer cells. (F) Heatmap of relative expression of genes filtered with $\text{Log}_2$ FC (acid/non-acid) > |0.5| and $P < 0.05$, that are involved in DNA repair (GO0006281). (G) PPI network analysis of the DE protein-coding genes in acid-exposed cancer cells using the Metascape MCODE algorithm to identify neighborhoods where proteins are densely connected. MCODE network nodes are displayed in red or blue color to represent upregulated or downregulated DEGs in the transcriptome of acid-exposed cancer cells, respectively. Data information: data are representative of $n = 3$ independent biological replicates. (F) $P$ values filtering was determined from statistics in Fig. 1E. Each column represents relative expression values in independent biological replicates.

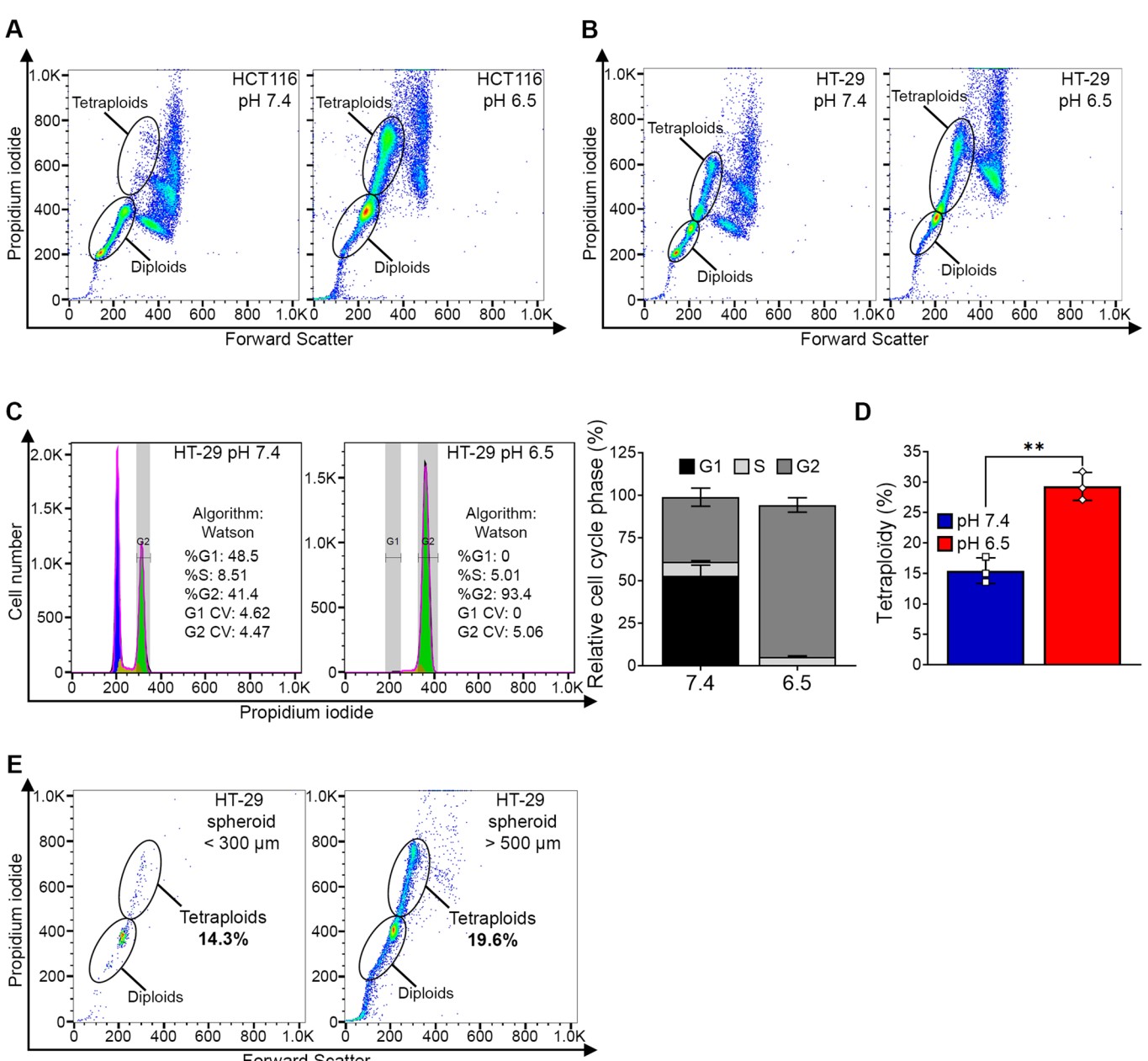

**Figure EV3. Acid-exposed cancer cells accumulate as tetraploid cells.**

(A, B) Representative propidium iodide (PI) signal *vs.* forward scatter (FSC) plots of flow cytometry analysis of HCT116 (A) or HT-29 (B) cells cultured at pH 6.5 (right panels) *vs.* pH 7.4 (left panels). Note that the areas of interest on the PI *vs.* FSC charts were established in cells at pH 7.4 (A, B, left panels) and do not intersect for diploid G2/M and tetraploid G1 cells; fluorescence intensity of PI may indeed fluctuate based on DNA packing according to the cell cycle phase. (C) Flow cytometry analysis of DNA content was used to determine cell cycle distribution of HT-29 cells cultured at pH 6.5 *vs.* pH 7.4. Representative HT-29 cell cycle analysis (C, left panels) and quantification (C, right bar graph) are shown. (D) Proportion of tetraploid HT-29 cancer cells determined as described in (B) from cells cultured at pH 6.5 or 7.4. (C–E) Representative PI signal *vs.* FSC plots of flow cytometry analysis from $n = 30$ independent dissociated HT-29 spheroids with size > 500 μm (right panel) *vs.* < 300 μm (left panel). Data information: data are representative of $n = 3$ independent biological replicates. (C, D) Quantification data are presented as means ± SD of $n = 3$ independent biological replicates. The relative proportion of HT-29 cells in G1, S and G2/M phases was determined using the Watson algorithm in (C). Statistical analysis was performed using an unpaired two-tailed Student's *t* test (**$P < 0.01$) in (D).

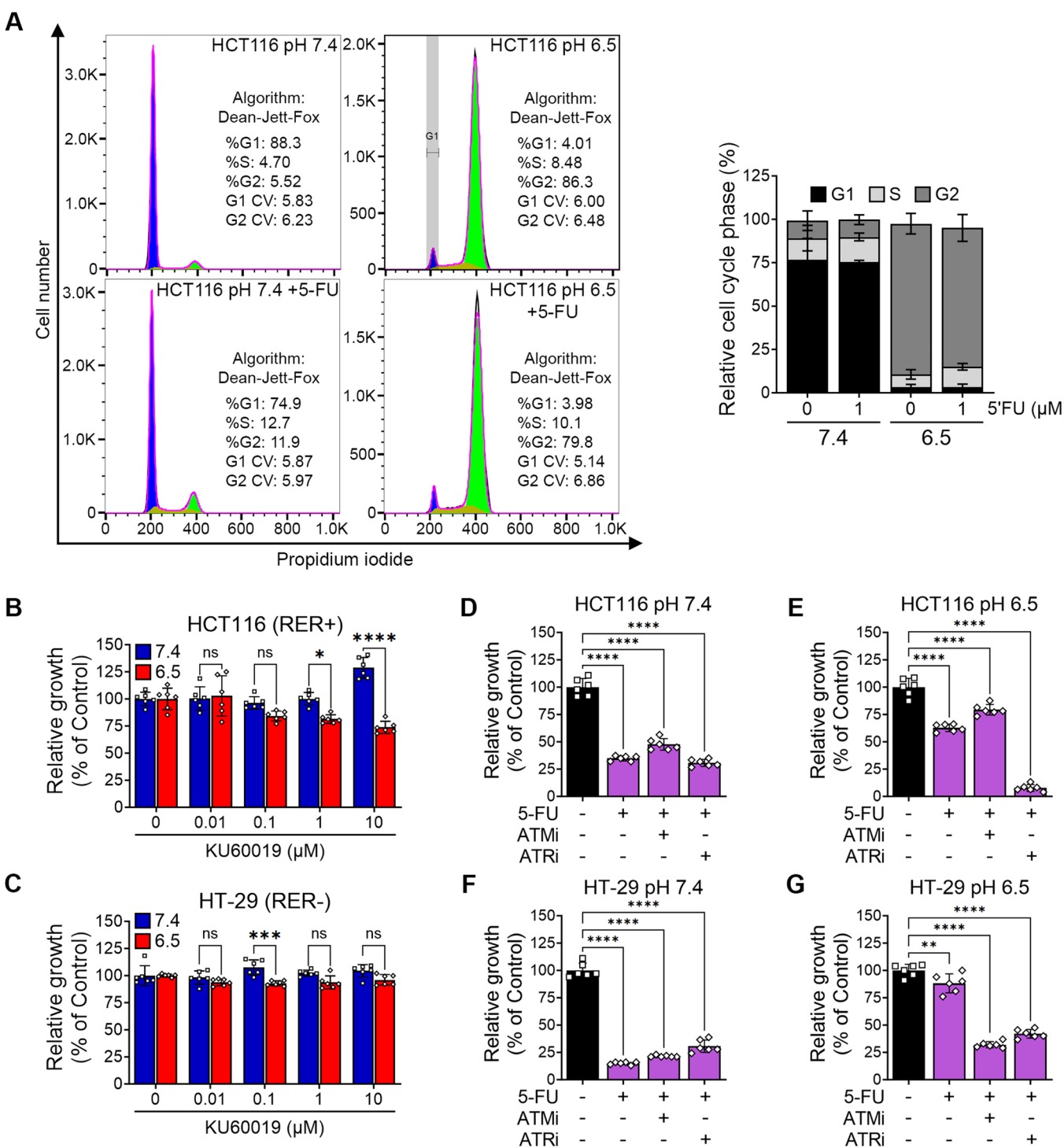

**Figure EV4.  The proportion of G2/M arrested acid-exposed cancer cells is not influenced by 5-FU exposure and growth of acid-exposed cancer cells is inhibited either by single ATMi KU60019 or combination of 5-FU with ATMi or ATRi.**

(A) Flow cytometry analysis of DNA content was used to determine cell cycle distribution of HCT116 cultured at pH 6.5 *vs.* pH 7.4 and exposed (or not) to 1 μM 5-FU. Representative cell cycle analysis plots (left panels), and quantification (right bar graph) are shown. (B–G) Cell viability assays in HCT116 (B, D, E) and HT-29 (C, F, G) cancer cells cultured at pH 7.4 or 6.5, and treated with the indicated dose of ATMi KU60019 (B, C), or 100 μM 5-FU alone or in combination with 1 μM ATMi AZD0156 or 0.1 μM ATRi Elimusertib (D–G) for 72 h. Data information: (A) The relative proportion of cells in G1, S and G2/M phases was determined using the Dean–Jett–Fox algorithm. Data are represented as means ± SD of n = 3 independent biological replicates. For panels (B–G) bar graphs represent means ± SD (six technical replicates, n = 3 independent biological replicates) and significance was determined using two-way ANOVA (B, C) or one-way ANOVA (D–G) with post hoc Tukey's multiple-comparison analysis (ns non significant; *$P < 0.05$; **$P < 0.01$; ***$P < 0.001$; ****$P < 0.0001$).

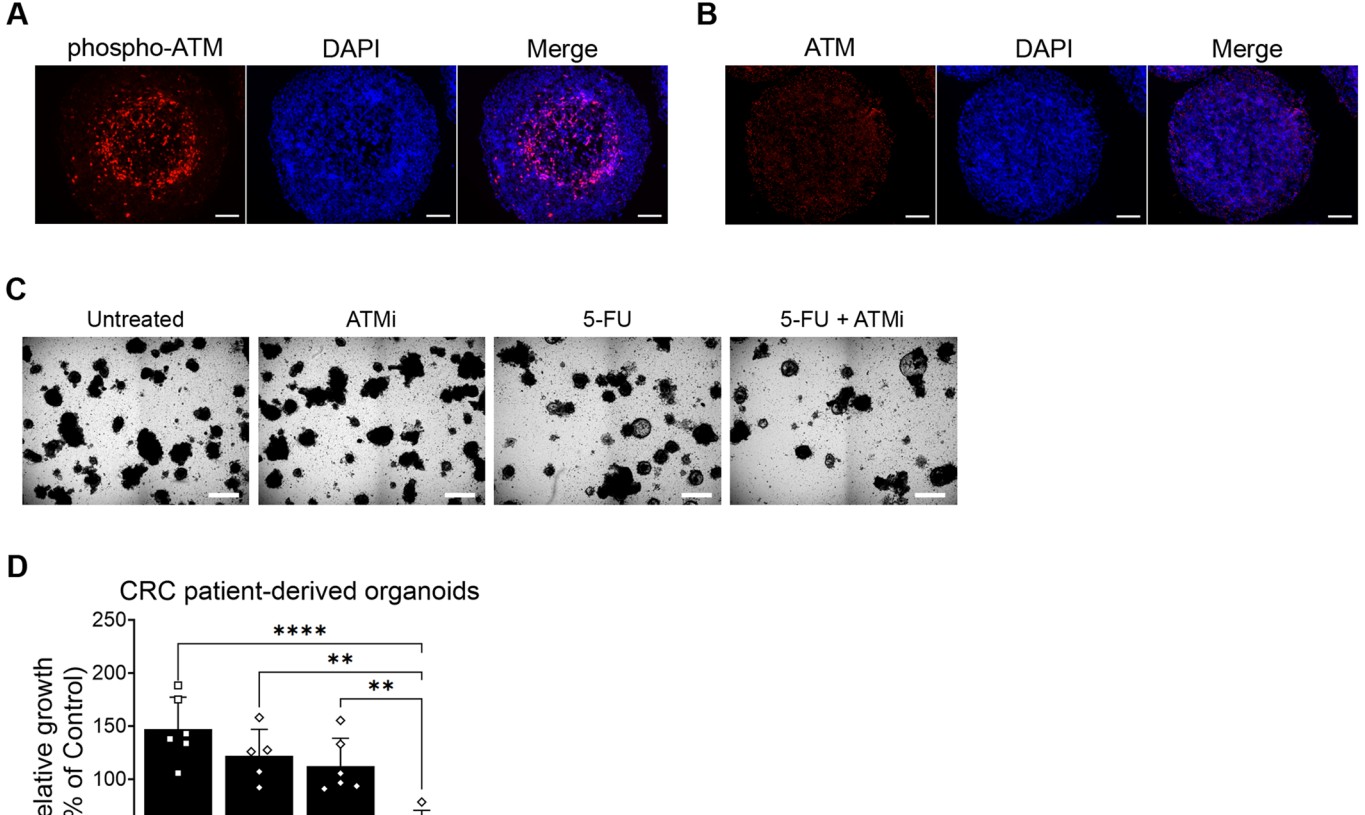

**Figure EV5. ATM is activated in the core of 3D spheroids and combination of 5-FU with ATMi results in growth inhibitory effects on patients-derived organoids.**

(A, B) Immunofluorescence labeling of 3D HCT116 spheroid equatorial sections with antibodies targeting either phospho-ATM (A) or ATM (B). Scale bars = 100 µm. (C, D) Effects of 10 µM ATMi AZD0156, 10 µM 5-FU or the combination of both drugs on the growth of colorectal cancer patient-derived tumor organoids. Representative pictures of organoids at day 7 post-treatment are presented (C) together with quantification (D). Scale bars = 500 µm. Data information: (D) Bar graph represents means ± SD of $n = 5$–6 independent biological replicates and significance was determined using one-way ANOVA with Tukey's multiple-comparison analysis (**$P < 0.01$; ****$P < 0.0001$). Source data are available online for this figure.

