## [Peer Review File · EMBO Reports]

Tumor acidosis-induced DNA damage response and tetraploidy enhance sensitivity to ATM and ATR inhibitors

Léo Aubert, Estelle Bastien, Ophélie Renoult, Céline Guilbaud, Kubra Ozkan, Davide Brusa, Caroline Bouzin, Elena Richiardone, Corentin Richard, Romain Boidot, Daniel Léonard, Cyril Corbet, and Olivier Feron

Corresponding author(s): Olivier Feron (olivier.feron@uclouvain.be), Léo Aubert (leo.aubert@uclouvain.be)

Review Timeline:

Submission Date:	8th Oct 23
Editorial Decision:	14th Nov 23
Revision Received:	5th Jan 24
Editorial Decision:	15th Jan 24
Revision Received:	27th Jan 24
Accepted:	29th Jan 24

Editor: Achim Breiling

Transaction Report:

Dear Prof. Feron,

Thank you for the submission of your research manuscript to EMBO reports. I have now received the reports from the three referees that were asked to evaluate your study, which can be found at the end of this email.

As you will see, the referees think that the findings are of interest. However, they have several comments, concerns, and suggestions, indicating that a major revision of the manuscript is necessary to allow publication of the study in EMBO reports. As the reports are below, and all the referee concerns need to be addressed, I will not detail them here.

Given the constructive referee comments, I would like to invite you to revise your manuscript with the understanding that all referee concerns must be addressed in the revised manuscript or in a detailed point-by-point response. Acceptance of your manuscript will depend on a positive outcome of a second round of review. It is EMBO reports policy to allow a single round of revision only and acceptance of the manuscript will therefore depend on the completeness of your responses included in the next, final version of the manuscript.

- 1) a .docx formatted version of the final manuscript text (including legends for main figures, EV figures and tables), but without the figures included. Figure legends should be compiled at the end of the manuscript text.
- 2) individual production quality figure files as .eps, .tif, .jpg (one file per figure), of main figures (up to 8) and EV figures. Please upload these as separate, individual files upon re-submission.

- 4) a complete author checklist, which you can download from our author guidelines (<https://www.embopress.org/page/journal/14693178/authorguide>). Please insert page numbers in the checklist to indicate where the requested information can be found in the manuscript. The completed author checklist will also be part of the RPF.

- 5) that primary datasets produced in this study (e.g. RNA-seq, ChIP-seq, structural and array data) are deposited in an

appropriate public database. If no primary datasets have been deposited, please also state this in a dedicated section (e.g. 'No primary datasets have been generated and deposited'), see below.

The accession numbers and database should be listed in a formal "Data Availability" section (placed after Materials & Methods) that follows the model below. This is now mandatory (like the COI statement). Please note that the Data Availability Section is restricted to new primary data that are part of this study. This section is mandatory. As indicated above, if no primary datasets have been deposited, please state this in this section

Data availability

8) Regarding data quantification and statistics, please make sure that the number "n" for how many independent experiments were performed, their nature (biological versus technical replicates), the bars and error bars (e.g. SEM, SD) and the test used to calculate p-values is indicated in the respective figure legends (also for potential EV figures and all those in the final Appendix). Please also check that all the p-values are explained in the legend, and that these fit to those shown in the figure. Please provide statistical testing where applicable. Please avoid the phrase 'independent experiment', but clearly state if these were biological or technical replicates. Please also indicate (e.g. with n.s.) if testing was performed, but the differences are not significant. In case n=2, please show the data as separate datapoints without error bars and statistics. See also: <http://www.embopress.org/page/journal/14693178/authorguide#statisticalanalysis>

9) Please add scale bars of similar style and thickness to all the microscopic images, using clearly visible black or white bars (depending on the background). Please place these in the lower right corner of the images themselves. Please do not write on or near the bars in the image but define the size in the respective figure legend.

10) Please also note our reference format:

12) We now use CRedit to specify the contributions of each author in the journal submission system. CRedit replaces the author contribution section. Please use the free text box to provide more detailed descriptions and do not provide your final manuscript text file with an author contributions section. See also our guide to authors: <https://www.embopress.org/page/journal/14693178/authorguide#authorshippinguidelines>

13) We would encourage you to use 'Structured Methods', our new Materials and Methods format. According to this format, the

Materials and Methods section should include a Reagents and Tools Table (listing key reagents, experimental models, software and relevant equipment and including their sources and relevant identifiers) followed by a Methods and Protocols section in which we encourage the authors to describe their methods using a step-by-step protocol format with bullet points, to facilitate the adoption of the methodologies across labs. More information on how to adhere to this format as well as downloadable templates (.doc or .xls) for the Reagents and Tools Table can be found in our author guidelines (section 'Structured Methods'):

14) Please reduce the number of keywords to 5 and order the manuscript sections like this, using these names:

Title page - Abstract - Keywords - Introduction - Results - Discussion - Materials and Methods - Data availability section - Acknowledgements - Disclosure and Competing Interests Statement - References - Figure legends - Expanded View Figure legends

I look forward to seeing a revised version of your manuscript when it is ready. Please let me know if you have questions or comments regarding the revision.

Yours sincerely,

Referee #1:

I have reviewed an earlier version of this manuscript submitted to a different journal, and indicated my support for it. This manuscript is much improved and addresses most of my comments, therefore I do not have many more to add. I supported its publication in the original journal and continue to do so with EMBO reports.

The work is from a top-tier laboratory with a strong track record. The work has innovation and presents technically challenging experimnts, that I trust the other reviewers will also appreciate.

In essence, the manuscript describes the differences in transcriptomic profile established in cells of distinct milieu pH, identified by pHLIP and sorted accordingly. The canonical view has held that many of these differences are dominated by metabolic rewiring which the authors confirm. Above and beyond, the novelty is finding DNA damage response as an enriched pathway. The finding of tetraploidy (Fig 3) is particularly interesting and novel. The work then moves toward human organoids to confirm findings with drugs and discuss wider impact.

The study is robust and interesting, and adds a new dimension to pH studies. The discussion around RER+ and RER- lines is important and compelling. The strength is the unbiased approach, innovative use of pHLIP-based sorting (which is technically very challenging). The power of the tests is high due to the paired nature, and should also be highlighted.

Comments:

1. I appreciate the naming convention on p 6 that acidic cell means cell exposed to low pH, but this may be lost to the reader and inferred as intracellular. Perhaps "acid-exposed cell" is better.
2. To what degree is pHLIP an indicator of an on-going acidosis? It inserts irreversibly and may indicate a history of acidosis, which could skew the interpretation?
3. Early results or even introduction should justify choice of HCT116 cells as a RER+ line.
4. The tetraploidy result is striking. Perhaps it could be useful to show individual results not just mean and error in Fig 3B to get a sense of spread.
5. Fig 4 describes RER- and RER+ lines -it would be useful to indicate which is which in figure labelling.
6. The results on the p53 are discussed but not shown; it would be important to include this, in my view, as a supplement.

Referee #2:

In this manuscript, the authors employed pH sensitive peptide pHLIP for the identification of acidic cancer cells in 3D culture. They used FACS to sort out acidic and non-acidic cells and performed RNA-seq analysis, which led the discovery of metabolic pathways as well as cell cycle/DNA damage pathways that were altered in acidic cells. The authors further validated their results using 2D cultures under normal versus acidic conditions. Additionally, the authors showed increased DNA damage signaling and sensitivity to ATMi/ATRi or 5-FU in combination with ATMi/ATRi.

The authors showed convincingly cell cycle was altered in 2D culture under acidic condition. According to their FACS data presented in Supplementary Figure 3 as well as in Figure 3, this may be largely due to the significant increase of tetraploid cells under this condition, but not due to G2/M arrest as speculated in this manuscript. These tetraploid cells may still undergo cell cycle progression with slightly increased DNA damage responses. Thus, these cells only showed modest sensitivity to ATMi/ATRi (Figure 4). Moreover, there was also modest sensitivity to 5-FU and ATMi/ATRi combinations shown in Figure 5. As a matter of fact, 2D cells under normal/non-acidic growth condition should also be sensitive to 5-FU plus ATMi/ATRi combinations.

Nevertheless, it is quite interesting that acidic condition led to drastic increase in tetraploid cells. The authors may want to define the mechanisms underlying the formation of tetraploid cells, which are likely due to failure of cytokinesis. It is not clear whether there are any agents that would specifically eliminate tetraploid cells, which may worth testing. Moreover, the authors should determine whether this increase in tetraploid cells could be observed in 3D culture.

Referee #3:

The authors completed a screen of acidic vs non-acidic cells in a human cancer cell line grown in spheroids. The acidic vs. non-acidic cells were separated using FACS on the basis of binding by pHLIP, but not K-pHLIP. The screen identified DDR genes upregulated in acidic cells. DDR signaling is upregulated in cells cultured in acidic conditions. ATM and ATR kinase inhibitors preferentially kill cells cultured in acidic conditions. ATM and ATR kinase inhibitors combined with 5-FU potentiate the preferential kill of cells cultured in acidic conditions.

The figures are of a high quality and the screen is validated, in part, by the monolayer tissue culture experiments. However, I do have concerns about the controls completed with the screen.

Concerns:

The authors need to clarify the use of pHLIP and K-pHLIP in their screen (Figure 1 and Supplemental Figure 1). The manuscript foundation is a screen that is entirely dependent on the binding of these peptides and the lack of clarity in the description, controls, and perhaps validation, of the binding of these peptides undermines the manuscript. My understanding from reading the submission is that pHLIP binds nonspecifically, without membrane insertion. Since K-pHLIP cannot insert, it reproduces the nonspecific binding, but not the membrane insertion and specific binding. I don't believe the authors do enough to document this fundamental aspect of their paper by showing one spheroid in Figure 1B. Perhaps more images would help, including higher magnification. Furthermore, is the staining of the center of the organoids with pHLIP uniform? Can this be shown, if the authors believe it's important.

It's unfortunate that this background staining couldn't be eliminated and I appreciate that the authors have gone to great lengths to try to remove this background contamination from their screen, but at the very least more precise language is needed to explain what they have done. This is illustrated by the line "since labeling of the central core of the spheroids excluded an issue of penetration depth, the rim staining could instead arise for non-specific binding at the medium/cell interface." Can the authors clarify how the issue of penetration depth may impact their data - what does this mean? The title of the second section of the results starts with "The Acidic Compartment.." The reader is asked to accept that Figure 1 shows beyond reasonable doubt that the acidic cells have been isolated from the non-acidic cells. I think more data is needed to support that claim. The analyses are excellent, the figures are beautiful, my issue is what is the analysis of beyond pHLIP+K-pHLIP- cells in spheroids? Can the screen be validated in spheroids by immunofluorescence for DDR signaling?

Can the authors clarify what "regardless of cell replication error status" means in the abstract.

The manuscript is interesting and potentially impactful. I think the above issues just need to be clarified.

Referee #1:

I have reviewed an earlier version of this manuscript submitted to a different journal, and indicated my support for it. This manuscript is much improved and addresses most of my comments, therefore I do not have many more to add. I supported its publication in the original journal and continue to do so with EMBO reports.

The work is from a top-tier laboratory with a strong track record. The work has innovation and presents technically challenging experiments, that I trust the other reviewers will also appreciate.

In essence, the manuscript describes the differences in transcriptomic profile established in cells of distinct milieu pH, identified by pHLIP and sorted accordingly. The canonical view has held that many of these differences are dominated by metabolic rewiring which the authors confirm. Above and beyond, the novelty is finding DNA damage response as an enriched pathway. The finding of tetraploidy (Fig 3) is particularly interesting and novel. The work then moves toward human organoids to confirm findings with drugs and discuss wider impact.

The study is robust and interesting, and adds a new dimension to pH studies. The discussion around RER+ and RER- lines is important and compelling. The strength is the unbiased approach, innovative use of pHLIP-based sorting (which is technically very challenging). The power of the tests is high due to the paired nature, and should also be highlighted.

We thank this Reviewer for the thoughtful and constructive comments.

Comments:

1. I appreciate the naming convention on p 6 that acidic cell means cell exposed to low pH, but this may be lost to the reader and inferred as intracellular. Perhaps "acid-exposed cell" is better.

The text was modified accordingly.

2. To what degree is pHLIP an indicator of an on-going acidosis? It inserts irreversibly and may indicate a history of acidosis, which could skew the interpretation?

The reversibility of the pHLIP anchoring into the membrane of acid-exposed cells was studied by the group of Donald Engelman (PMID: 32409607). At acidic pHe, the loss of charge and increase in overall hydrophobicity drive pHLIP peptide to partition across the hydrophobic core of the membrane bilayer to form transmembrane helix. This helix spans the lipid bilayer, placing the C terminus in the intracellular space where, due to the more alkaline pH in the cytosol, it becomes deprotonated and charged, stably anchoring the peptide in the cell membrane. Still, Slaybaugh et al. (PNAS 2020) documented that while most, if not all, protonable residues of the pHLIP peptide need to be protonated (ie, become neutral) to enter the lipid bilayer, only one Asp deprotonation is enough to destabilize helix and promote rapid peptide exit. These data support the potential reversibility of pHLIP peptide insertion when pHe is back to more neutral values. Besides these biophysical considerations, in the absence of dynamic blood flow, it is reasonable to say that acidosis follows the growth of 3D spheroids

(ie, acidic area enlarges with spheroid size – see our previous work, Corbet et al., Nature Comm, PMID: 31974393) so that fluorophore-conjugated pHLIP peptide insertion label cancer cells facing ongoing acidic pH. Also, the short incubation time (24h) with pHLIP peptide is unlikely to be associated with major pH fluctuations in 3D spheroids.

3. Early results or even introduction should justify choice of HCT116 cells as a RER+ line.

HCT116 cell model was originally chosen for the ability of these CRC cells to form highly reproducible spheroids largely deprived of a necrotic zone prone to interfere with pHLIP labelling. We were not anticipating DDR as a major acidosis-dependent pathway at the beginning of this work. Still, we have now better emphasized the concept of replication error deficiency (RER) issue after reporting the tetraploidy observation in RER-positive HCT116 cells (Figure 3). Accordingly, HT29 CRC cells that are RER-negative allowed to document that acidosis-induced increased DDR response and tetraploidy could not be directly related to an intrinsic mismatch repair deficiency.

4. The tetraploidy result is striking. Perhaps it could be useful to show individual results not just mean and error in Fig 3B to get a sense of spread.

Figures 3B and EV3D were adapted to show individual data sets.

5. Fig 4 describes RER- and RER+ lines -it would be useful to indicate which is which in figure labelling.

The figure was adapted to indicate the RER status.

6. The results on the p53 are discussed but not shown; it would be important to include this, in my view, as a supplement.

p53 data are now presented as Appendix Figures S1E-F.

Referee #2:

In this manuscript, the authors employed pH sensitive peptide pHLIP for the identification of acidic cancer cells in 3D culture. They used FACS to sort out acidic and non-acidic cells and performed RNA-seq analysis, which led the discovery of metabolic pathways as well as cell cycle/DNA damage pathways that were altered in acidic cells. The authors further validated their results using 2D cultures under normal versus acidic conditions. Additionally, the authors showed increased DNA damage signaling and sensitivity to ATMi/ATRi or 5-FU in combination with ATMi/ATRi.

The authors showed convincingly cell cycle was altered in 2D culture under acidic condition. According to their FACS data presented in Supplementary Figure 3 as well as in Figure 3, this may be largely due to the significant increase of tetraploid cells under this condition, but not due to G2/M arrest as speculated in this manuscript. These tetraploid cells may still undergo cell cycle progression with slightly increased DNA damage responses.

We agree with the Reviewer that diploid G2/M and tetraploid G1 phase cancer cells share a theoretical 4N DNA content. We have acknowledged this caveat in the Results section and now refer to indirect evidence to support G2/M arrest. Indeed, in Fig. 2F, we report that GSEA reveals significant increase of the G2/M pathway in acid-exposed cancer cells (NES= 3.78, $p < 0.001$), highlighting a link between acid-upregulated transcripts and DNA damage checkpoints. Also, regions of interest in the propidium iodide (PI) signal vs. FSC charts (determined in cells at pH 7.4, left panels of Figs. EV3A and EV3B) do not intersect for diploid G2/M and tetraploid G1 phases in acid-exposed cancer cells (pH 6.5). Biological explanation could be that propidium iodide (PI)-based DNA staining is nonlinear since fluorescence intensity may vary depending on the cell cycle phase-dependent DNA packing and according to binding saturation at high DNA concentration.

Thus, these cells only showed modest sensitivity to ATMi/ATRi (Figure 4). Moreover, there was also modest sensitivity to 5-FU and ATMi/ATRi combinations shown in Figure 5. As a matter of fact, 2D cells under normal/non-acidic growth condition should also be sensitive to 5-FU plus ATMi/ATRi combinations.

We agree with the Reviewer that the effects of ATMi/ATRi as single treatment (Fig. 4G-J) may look modest (although statistically significant) but the combination with 5FU demonstrates a substantial advantage compared to 5-FU alone (Fig. 5). Some of the latter 3D data, in particular those related to HT-29 spheroids (see first 5 bars in Figure 5C-D), suggested that the increased DDR response under acidosis could in part account for resistance to 5-FU (in the acidic tumor compartment). We now provide a new set of data using 2D cultures of HCT116 and HT-29 cancer cells (new Figures EV4D-G) showing that both acid-exposed cancer cell types are more resistant to 5-FU (than corresponding cancer cells maintained at pH 7.4) but exhibit an enhanced growth inhibitory response in the presence of ATRi (and ATMi for acid-exposed HT-29 cancer cells).

Nevertheless, it is quite interesting that acidic condition led to drastic increase in tetraploid cells. The authors may want to define the mechanisms underlying the formation of tetraploid

cells, which are likely due to failure of cytokinesis. It is not clear whether there are any agents that would specifically eliminate tetraploid cells, which may worth testing.

We have now briefly commented on the origin of tetraploidy or whole genome doubling (WGD) including cytokinesis failure but also endoreplication. About compounds targeting tetraploid cells, there are only a few reports describing effects of such drugs and most of them usually suffer from a deficit in selectivity confusing any interpretation of data. We explored the effects of one of them, an inhibitor of POLO-like 1 kinase (PLK-1) that we identified among the enriched pathway in acid-exposed cancer cells (Figures 2A-B) and which is known to support mitotic entry following recovery from DNA damage in polyploid cells (PMID: 15350223). Our preliminary exploration of BI-2536, a PLK-1 inhibitor, revealed that while the growth inhibitory effects were similar in CRC cells kept at pH 7.4 compared to those exposed to pH 6.5 (even less pronounced for the latter), the reappearance of cell growth following drug removal was notably reduced in the presence of acidosis (Appendix Figure S2). These findings support previous observations of polyploid cells being more readily moved toward mitotic catastrophe-induced apoptosis upon PLK-1 inhibition or silencing (PMID: 32259417). Further experiments are warranted to determine the safety of such approaches and whether other therapeutic strategies targeting tetraploid cells may be more appropriate.

Moreover, the authors should determine whether this increase in tetraploid cells could be observed in 3D culture.

To abide by the Reviewer's suggestion, we took advantage of acidosis developing in proportion to the size of 3D tumor spheroids as we previously reported (Corbet et al., Nature Comm 2020) and documented an increase in the amount of tetraploid cells in HT-29 spheroids with a diameter >500 μm (vs. those with a diameter <300 μm) (new Figure EV3E).

Referee #3:

The authors completed a screen of acidic vs non-acidic cells in a human cancer cell line grown in spheroids. The acidic vs. non-acidic cells were separated using FACS on the basis of binding by pHLIP, but not K-pHLIP. The screen identified DDR genes upregulated in acidic cells. DDR signaling is upregulated in cells cultured in acidic conditions. ATM and ATR kinase inhibitors preferentially kill cells cultured in acidic conditions. ATM and ATR kinase inhibitors combined with 5-FU potentiate the preferential kill of cells cultured in acidic conditions.

The figures are of a high quality and the screen is validated, in part, by the monolayer tissue culture experiments. However, I do have concerns about the controls completed with the screen.

Concerns:

The authors need to clarify the use of pHLIP and K-pHLIP in their screen (Figure 1 and Supplemental Figure 1). The manuscript foundation is a screen that is entirely dependent on the binding of these peptides and the lack of clarity in the description, controls, and perhaps validation, of the binding of these peptides undermines the manuscript. My understanding from reading the submission is that pHLIP binds nonspecifically, without membrane insertion. Since K-pHLIP cannot insert, it reproduces the nonspecific binding, but not the membrane insertion and specific binding. I don't believe the authors do enough to document this fundamental aspect of their paper by showing one spheroid in Figure 1B. Perhaps more images would help, including higher magnification. Furthermore, is the staining of the center of the organoids with pHLIP uniform? Can this be shown, if the authors believe it's important.

We agree with the Reviewer that our description of the mechanism of action of pHLIP peptide was a bit clumsy. To make it clear, we could have used binary pHLIP-positive and pHLIP-negative FACS sorting to isolate acidic and non-acidic cancer cells from 3D spheroids. However, we thought that including a control peptide could make this sorting more robust and specific because it could help to get rid of potential unspecific labelling of the pHLIP peptide. We used the most adequate control peptide for this purpose, i.e. a K-pHLIP peptide wherein protonatable aspartate residues are replaced by positively charged lysine residues so that it loses its ability to fold and integrate into plasma membranes of cells exposed to acidic extracellular pH. FACS sorting eventually identified double positive (pHLIP+ and K-pHLIP+) cell populations proving that this control was indeed necessary. This led us to exclude double positive cells for further RNA seq studies of acidic cancer cells.

Fluorescent peptide photomicrographs were included in Figure 1B to illustrate the most likely explanation of this double labelling: an unspecific (ie, non pH-dependent) labelling of the external cell layer of 3D spheroids. We have now added a panel in Figure 1B depicting fluorescence signal from either peptide. It shows a similar signal extent of pHLIP and K-pHLIP peptides at a depth <50µm (ie, the external rim of spheroids), this level was higher than the detected signal in the underlying cell layers (between 50 and 100 µm) but much lower than the central acidic core of the spheroids only labelled by pHLIP peptide. We have also more explicitly described the fundamentals of the pHLIP technology in the revised version of our manuscript.

It's unfortunate that this background staining couldn't be eliminated and I appreciate that the authors have gone to great lengths to try to remove this background contamination from their screen, but at the very least more precise language is needed to explain what they have done. This is illustrated by the line "since labeling of the central core of the spheroids excluded an issue of penetration depth, the rim staining could instead arise for non-specific binding at the medium/cell interface." Can the authors clarify how the issue of penetration depth may impact their data - what does this mean?

We agree that the wording about penetration depth was unclear. In addition of our comments above, we have rewritten this paragraph to better explain the issue of the rim staining: "The peripheral staining at the spheroid/medium interface evoked unspecific labelling due to membrane adsorption without the expected coil-helix transition and membrane integration occurring at low pH_e. To prove this issue, we used a K-pHLIP peptide wherein protonatable aspartate residues were replaced by positively charged lysine residues so that it loses its ability to fold and integrate into plasma membranes of cells exposed to acidic extracellular pH (17). This K-pHLIP control experiment confirmed the unspecific (ie, non pH-dependent) peptide labelling of the external layers of 3D spheroids (that are in close contact with the buffered culture medium) (Figure 1B, right panel)."

The title of the second section of the results starts with "The Acidic Compartment.." The reader is asked to accept that Figure 1 shows beyond reasonable doubt that the acidic cells have been isolated from the non-acidic cells. I think more data is needed to support that claim. The analyses are excellent, the figures are beautiful, my issue is what is the analysis of beyond pHLIP+ K-pHLIP- cells in spheroids? Can the screen be validated in spheroids by immunofluorescence for DDR signaling?

As a first response to this comment, we have now better emphasized that pHLIP^{pos}/K-pHLIP^{neg} cells can be aligned with the acidic tumor compartment in regard of the existing literature about tumor acidosis and our own independent data sets generated from acid pH-adapted cancer cells:

- GSEA carried out on pHLIP⁺/K-pHLIP⁻ cells identified oxidative phosphorylation (NES=3.09, P<0.001), fatty acid metabolism (NES = 1.52, P<0.01) and reactive oxygen species (NES = 1.82, P<0.01) pathways (Figure 1F), which have been previously identified as specific features of cancer cells exposed to acidic pH_e (PMID: 32764426; 35263578; 36533672; 27508876).

- DDR pathway enrichment in the pHLIP⁺/K-pHLIP⁻ cells isolated from 3D spheroids is confirmed by the stimulated DDR signaling in different acid-adapted cancer cells (pH 6.5) (vs. cells maintained at neutral pH 7.4) (Figures 3 and 4).

To adhere more closely to the Reviewer's suggestion, we now also provide immunofluorescence data confirming the presence of phospho-ATM in the central region of the spheroids (compatible with the acidic compartment) but not in the external cell layers (Figure EV5A).

Can the authors clarify what "regardless of cell replication error status" means in the abstract.

HCT116 and HT-29 have distinct replication error status. HCT116 is a CRC cell line that is DNA replication error (RER)-positive because of the lack of hMHL1 expression. RER-positive cancer cells are characterized by defects in the mismatch-repair system that allow frameshift mutations to accumulate (probably because of DNA polymerase slippage during replication). Approximately 15% of sporadic colorectal cancers (CRCs) are microsatellite unstable (MSI+) so that HCT116 cells do not represent the majority of CRC cells. Therefore, the use of acid-exposed HT-29 cells that are RER-negative was critical to prove that acidosis-induced increased DDR response and tetraploidy were beyond intrinsic mismatch repair deficiency, thereby enlarging the bearing of our findings.

The text was modified to integrate this comment.

The manuscript is interesting and potentially impactful. I think the above issues just need to be clarified.

We thank the Reviewer for his/her valuable comments and agree that the requested clarifications enhance the biological relevance and understanding of our study.

++++

Dear Prof. Feron,

Thank you for the submission of your revised manuscript to our editorial offices. I have now received the reports from the three referees that I asked to re-evaluate your study, you will find below. As you will see, the referees now fully support the publication of the study in EMBO reports. Referee #2 has a final suggestions to improve the manuscript, I ask you to address in a final revised manuscript.

- Please provide the abstract written in present tense throughout.
- Please upload the final figure files (main and EV figures) without their legends. The figure legends should be provided only in the main manuscript text file.
- Please check these name discrepancies: 'Elena Richiardone' on the title page of the manuscript text file vs. 'Elena Richiardione' in the submission system; 'Corentin Richard' on the title page of the manuscript text file vs. RICHARD Corentin in the submission system.
- Please upload the dataset as Excel file and put the legend into in the file on the first TAB. Please label and name the dataset "Dataset EV1" (not "Suppl. Table S1") and add a callout for Dataset EV1 to the main manuscript text.
- The Data Availability section should only contain information on large datasets that have been deposited to external repositories and all access information. Please remove the statement: 'Further information and requests for resources and reagents should be directed to and will be fulfilled by the Lead Contact, Prof. Olivier Feron (olivier.feron@uclouvain.be). All unique reagents generated in this study will be made available on request by the Lead Contact with a completed material transfer agreement (MTA).' Please remove the referee token from the data availability section and make sure that the dataset is public latest upon online publication of the study.
- Please make sure that the number "n" for how many independent experiments were performed, their nature (biological versus technical replicates), the bars and error bars (e.g. SEM, SD) and the test used to calculate p-values is indicated in the respective figure legends (for main, EV and Appendix figures) of the final revised manuscript. Please also check that all the p-values are explained in the legend, and that these fit to those shown in the figure. Please provide statistical testing where applicable. Please avoid the phrase 'independent experiment', but clearly state if these were biological or technical replicates. Please also indicate (e.g. with n.s.) if testing was performed, but the differences are not significant. In case n=2, please show the data as separate datapoints without error bars and statistics. See also:

<http://www.embopress.org/page/journal/14693178/authorguide#statisticalanalysis>

If $n < 5$, please show single datapoints for diagrams. Could statistics also be added to the diagrams in Fig. 5 and in the Appendix? Moreover:

- Please note that the legends for figures 3d-f are not provided in the sequential manner (legend for figures 3e-f is provided before legend of figure 3d). This needs to be rectified.
- Please indicate the statistical test used for data analysis in the legends of figures 1e-f; 2d, f; EV 1e; EV 2f.
- Please note that information related to n is missing in the legends of figures 1e; 3a, c-d; EV 3c.
- Although 'n' is provided, please describe the nature of entity for 'n' in the legend of figure EV 5d.
- Please note that the error bars are not defined in the legends of figure 3a; EV 3c.
- Please note that the measure of center for the error bars needs to be defined in the legends of figures 5a-d.
- Moreover, please add to each legend a 'Data Information' section explaining the statistics used or providing information regarding replicates and scales. See:

- Please remove the reagents table from the main manuscript text file. I have attached templates for that in word or excel format. Please upload the filled in table to the manuscript tracking system as 'Reagent Table' file. Please also adjust any callouts to this table. The example linked below shows how the table will display in the published article and includes examples of the type of information that should be provided for the different categories of reagents and tools. Please list your reagents/tools using the categories provided in the template and do not add additional subheadings to the table. Reagents/tools that do not fit in any of the specific categories can be listed under "Other":

https://www.embopress.org/pb%2Dassets/embo-site/msb_177951_sample_FINAL.pdf

In addition, I would need from you:

Best,

Referee #1:

No further concerns. Fully support publication

Referee #2:

The authors addressed my previous concerns. I support the publication of this manuscript. However, I would encourage the authors to emphasize on tetraploidy but not DNA damage response, since tetraploidy appears to be the major feature in these acid exposed cancer cells.

Referee #3:

The authors have addressed my concerns thoroughly.

Referee #2:

The authors addressed my previous concerns. I support the publication of this manuscript. However, I would encourage the authors to emphasize on tetraploidy but not DNA damage response, since tetraploidy appears to be the major feature in these acid exposed cancer cells.

We have now modified the text so that we referred to tetraploidy induction in acid-exposed cancer cells as a main discovery of our study in the title of our manuscript, the abstract (line 4), the end of the introduction (line 3 of the last paragraph) and the first line of the discussion.

Prof. Olivier Feron
UCLouvain
IREC
57 Avenue Hippocrate B1.57.04
Brussels, Brussels 1200
Belgium

Dear Prof. Feron,

I am very pleased to accept your manuscript for publication in the next available issue of EMBO reports. Thank you for your contribution to our journal.

Yours sincerely,
